# Anthropocene climate warming enhances autochthonous carbon cycling in an upland Arctic lake, Disko Island, West Greenland

Mark A. Stevenson[1,2*], Suzanne McGowan[1], Emma J. Pearson[3], George E.A. Swann[1], Melanie J. Leng[4,5], Vivienne. J. Jones[6], Joseph J. Bailey[1,7#], Xianyu Huang[8], Erika Whiteford[9,10$]

[1]Centre for Environmental Geochemistry, School of Geography, University of Nottingham, University Park, Nottingham, NG7 2RD, UK

[2]School of Natural and Environmental Sciences, Newcastle University, Newcastle-upon-Tyne, NE1 7RU, UK

[3]School of Geography, Politics and Sociology, Newcastle University, Newcastle-upon-Tyne, NE1 7RU, UK

[4]National Environmental Isotope Facility, British Geological Survey, Keyworth, Nottingham, NG12 5GG, UK

[5]Centre for Environmental Geochemistry, School of Biosciences, University of Nottingham, Sutton Bonington Campus, Leicestershire, LE12 5RD, UK

[6]Environmental Change Research Centre, Department of Geography, University College London, London, WC1E 6BT, UK

[7]Geography Department, York St John University, YO31 7EX, UK

[8]State Key Laboratory of Biogeology and Environmental Geology and School of Geography and Information Engineering, China University of Geosciences, Wuhan 430078, China

[9]Department of Geography, Loughborough University, Loughborough, LE11 3TU, UK

[10]School of Science and Technology, Nottingham Trent University, Nottingham, NG11 8NS, UK

*Present address: School of Natural and Environmental Sciences, Newcastle University, Newcastle-upon-Tyne, NE1 7RU, UK

#Present address: Geography Department, York St John University, YO31 7EX, UK

$Present address: School of Science and Technology, Nottingham Trent University, Nottingham, NG11 8NS, UK

*Correspondence to*: Mark A. Stevenson (mark.stevenson@newcastle.ac.uk)

**Abstract.** The Arctic is rapidly changing, disrupting biogeochemical cycles and the processing, delivery and sedimentation of carbon (C), in linked terrestrial-aquatic systems. In this investigation, we coupled a hydrogeomorphic assessment of catchment soils, sediments and plants with a recent lake sediment sequence to understand the source and quality of organic carbon present in three Arctic upland lake catchments on Disko Island, located just south of the Low-High Arctic transition zone. This varied permafrost landscape has exposed soils with less vegetation cover at higher altitudes, and lakes received varying amounts of glacial meltwater inputs. We provide improved isotope and biomarker source identifications for palaeolimnological studies in high latitude regions, where terrestrial vegetation is at or close to its northerly and altitudinal range limit. The poorly developed catchment soils lead to lake waters with low dissolved organic carbon (DOC) concentrations ($\leq 1.5$ mg $L^{-1}$). Sedimentary Carbon/Nitrogen (C/N) ratios, the C isotope composition of organic matter ($\delta^{13}C_{org}$) and biomarker ratios (*n*-alkanes, *n*-alkanols, *n*-alkanoic acids and sterols) showed that sedimentary organic matter (OM) in these lakes is mostly derived from aquatic sources (algae and macrophytes). We used a $^{210}Pb$ dated sediment core to determine how carbon cycling in a lake-catchment system (Disko 2) had changed over recent centuries. Recent warming since the end of the Little Ice Age (LIA ~1860 AD), which accelerated after ca. 1950, led to melt of glacier ice and permafrost releasing nutrients and DOC to the lake, stimulating pronounced aquatic algal production, as shown by a > 10 fold increase in β-carotene, indicative of a major regime shift. We also demonstrate that recent increases in catchment terrestrial vegetation cover contributed to the autochthonous response. Our findings highlight that in Arctic lakes with sparsely developed catchment vegetation and soils, recent Anthropocene warming results in pronounced changes to in-lake C processing and the deposition of more reactive, predominately autochthonous C, when compared with extensively vegetated low Arctic systems.

**Keywords.** carbon cycling; lipid biomarkers; carbon isotopes; hydrogeomorphology; wetlands; lakes; sediments; plants; permafrost; Greenland; Arctic

## 1 Introduction

The carbon (C) stored and cycled in permafrost, soils and lake sediments of Arctic landscapes is a vital component of the terrestrial C budget (Hugelius et al., 2014; Anderson et al., 2019), but Arctic ecosystems are changing rapidly (Saros et al., 2019). Climate warming in Arctic regions (Smol and Douglas, 2007) can change the ecosystem structure of catchments and lakes with probable implications for landscape-scale C cycling (Anderson et al., 2018). In particular, changes in vegetation quantity and quality associated with long-term climate shifts can influence soil development (Wookey et al., 2009) and C transfer between terrestrial and aquatic environments. "Arctic greening", where warmer temperatures have enhanced terrestrial vegetation growth (Arndt et al., 2019) or "Arctic browning" where changing climate regimes such as drought and winter warming can reduce vegetation productivity (Phoenix and Bjerke, 2016), have the potential to alter primary production in lakes (McGowan et al., 2018), with associated implications for C cycling. Warming also leads to glacial recession increasing meltwater discharge into Arctic watersheds (Slemmons et al., 2013). Arctic regions lying close to the High Arctic - Low Arctic transition zone (below 75° N latitude in western Greenland (Daniels and De Molenaar, 1993)) are likely to experience particularly marked shifts in C cycling. Characterised by the presence (Low Arctic) or absence (High Arctic) of vegetation, warming may push marginal ecosystems into different ecological states (Corell et al., 2013).

Accurate methods for tracing how the quality and transfer of C is changing across catchment-lake systems are required for predicting broader scale future impacts on the Arctic C cycle (with global implications), for which geochemical analyses of lake-catchment samples are especially insightful (Karlsson et al., 2009; Lapierre and del Giorgio, 2012; McGowan et al., 2018). Previous studies in Greenland have used $\delta^{13}C_{org}$ and C/N to track particulate catchment inputs (Anderson et al., 2018; Leng et al., 2012), but due to their narrow ranges with some specimens, using these techniques in isolation cannot always distinguish source identities (Lacey et al., 2018; Holtvoeth et al., 2016). Here we additionally use lipid biomarker chain length compositions and the C isotopic composition of fatty acid methyl esters (FAMEs) as a potential solution to this challenge (Castañeda and Schouten, 2011). These techniques, which have been rarely applied in high latitude lake catchments, can provide additional understanding because shorter chains are typically indicative of algae and photosynthetic bacteria, mid-chains for macrophytes and longer chains of terrestrial plant input (Cranwell et al., 1987; Meyers, 2003; Ficken et al., 2000), while pigments (e.g. β-carotene) are effective autotroph biomarkers (McGowan, 2013). The type and quality of terrestrial OC supplied to lakes changes microbial communities and may ultimately regulate whether lakes are $CO_2$ sources or sinks (Jansson et al., 2000; Kortelainen et al., 2013).

Palaeoenvironmental records from lakes integrate carbon from the lake and catchment and allow the reconstruction of variability of C cycling on timescales that would otherwise be unobtainable (Leavitt et al., 1989; Leavitt et al., 2009). Palaeolimnology of Arctic lakes can be used to help estimate C burial rates over the Holocene (Anderson et al., 2009; Anderson et al., 2019) and to understand how lakes in different regions process terrestrial OM (McGowan et al., 2018). Investigations of carbon cycling across the transition between the cooler Little Ice Age (LIA) and warmer recent conditions in the Arctic provide a way of understanding how Arctic systems respond to climate change. Lacustrine records from close to the Jakobshavn Isfjord, near Ilulissat, east of Disko suggest

that the lowest Holocene temperatures occurred here in the 19[th] century (LIA) (Axford et al., 2013; Briner et al., 2016), rising in the early 20[th] century (Box et al., 2009; Yamanouchi, 2011) and accelerating with recent warming (Hanna et al., 2012).

Qeqertarsuaq (Disko Island, West Greenland) is situated in a Low Arctic region, just south of the High Arctic transition zone, and has a heterogeneous cover of herbaceous plants and bryophytes and several glaciers. This paper investigates the organic geochemistry of OM in the catchments and lake sediments on Disko Island to identify how terrestrial and aquatic $\delta^{13}C_{org}$ biomarkers (C/N ratios and $\delta^{13}C_{org}/\delta^{13}C_{FAMEs,}$ chlorophyll and carotenoid pigments) can be interpreted as palaeoenvironmental proxies in this sparsely vegetated Arctic region. By investigating sites spanning different catchment sizes, vegetation cover, and proximity to glacial features we aim to understand how these terrestrial factors influenced lake water quality and OC composition to ultimately be archived in the sediment record. We use this information to guide the interpretation of a sediment sequence from one lake spanning the transition from the LIA towards warmer conditions after 1950 to help understand how pronounced climate warming can alter terrestrial-aquatic carbon cycling.

## 2 Materials and methods

### 2.1 Study area

Qeqertarsuaq (Disko Island) (Fig. 1) is located within Disko Bugt (Bay), a large marine embayment between 69º15'N - 70º20'N and 51º50'W - 55º00'W. Disko is the largest island to the west of Greenland (8,575 km$^2$) and is extensively glaciated with main Sermersuaq (Storbræ) and Bræpasset ice caps covering the central plateaus. The geology is composed of 5,000 m deep and 60 M year old Tertiary basalts, part of the Disko-Nussuaq surge cluster (Chalmers et al., 1999; Humlum, 1996). There are prominent U-shaped valleys, cirques, talus slopes and fjords. Vegetation at a low altitude includes verdant dwarf shrub heaths in well-drained sites, and small mires in wet or snowpack-fed localities (Bennike, 1995). *Salix glauca* is the dominant shrub and sparser fell-field vegetation is present at higher elevations (Callaghan et al., 2011).

Disko Island has been subject to recent climate change, with a warming trajectory since the end of the LIA and at an increasing rate since the 1990s (Box, 2002; Hansen et al., 2006). This has been confirmed locally by tree ring evidence of winter warming (Hollesen et al., 2015) and both locally and regionally by instrumental records (Hanna et al., 2012; Hansen et al., 2006). Mechanistically, warming has contributed to the destabilisation of permafrost soils (Rowland et al., 2010) and retreat of catchment ice (including glaciers) (Anderson et al., 2017), releasing nutrients into downstream lakes. Old C released from melting catchment permafrost (Pautler et al., 2010; Schuur et al., 2009) could be one such source of C entering the lakes (as Disko is in a known permafrost active layer monitoring zone (Humlum et al., 1999)), along with minor contributions of catchment bryophytes and plant vegetation, evidenced by lipid ratios contributing to sedimentary C.

### 2.1 Hydro-geomorphological survey and catchment sampling

Three lakes (named Disko 1, 2 & 4), spanning a range of altitudes (214m to 575m) and catchment sizes (358 ha to 1,829 ha) were visited in April and August 2013 (Fig. 1c). In April, catchment soils and plants were sampled from exposed snow-free areas using a trowel into plastic bags. Further sampling was conducted in August 2013

to collect additional catchment soils, plants, benthic rock-scrape algae and lake edge water samples (lake edge, ~3-5 m) for nutrient, DOC and pigment analysis.

### 2.1.1 Vegetation composition surveys

Field notes, aerial imagery (LANDSAT & QuickBird through Google Earth) and mapping (Disko Island, Qeqertarsuaq 1:100,000 Hiking Map) were combined to estimate the percentage cover of alpine vegetation, bare
ground, snow cover and glaciers present in each lake catchment (including defining catchment boundaries) by manually delineating the areas using the 2D area function in Google Earth Pro 7.  In each lake catchment up to five 10x10 m plots were surveyed for vegetation cover, within which five 1x1 m quadrats were placed randomly, photographed and the percentage cover of plant functional types (defined as: total moss/lichen, total plants & total bare ground) were estimated, with key plant species identified following Rune (2011).

### 2.1.2 Water chemistry analysis

Water samples were filtered in the field using Whatman G/C filter papers, refrigerated and taken back to the laboratory for analysis. Filter papers were wrapped in foil, sealed in polythene and stored at −20ºC for chlorophyll and carotenoid (pigment) and trichromatic chlorophyll *a* analyses. TP (total phosphorus), SRP (soluble reactive phosphorus) and $NH_4$-N (ammonium) concentrations were measured by colorimetry using the methods in
Mackereth et al. (1978). DOC was measured using a Shimadzu TOC-VCSN analyser, with DOC calculated as the sum of organically bound C present in water that originate from compounds which can pass through a 0.45 µm filter. Total nitrogen (TN) was additionally analysed for the summer 2013 samples using a Ganimede TN analyser by alkaline digestion with peroxodisulphate.

### 2.2 Sediment core collection and chronology

In April 2013 sediments were sampled from Disko 2 and Disko 4 (Table 1) by drilling through lake ice using a HON-Kajak corer. Disko 2 was selected for intensive study due to its well-dated sediment sequence. Sediment samples were extruded at 0.5 cm intervals and water samples were taken under the lake ice at a depth of 1 m. The core (Disko 2 K1; length of core 31.5 cm) was analysed for $^{210}Pb$, $^{226}Ra$, $^{137}Cs$ and $^{241}Am$ by direct gamma assay using an ORTEC HPGe GWL Series well-type coaxial low background germanium detector at the Environmental
Change Research Centre, University College London. Total $^{210}Pb$ was measured by gamma emissions at 46.5keV, with $^{226}Ra$ by the combined daughter isotope emissions of 295keV and 325keV to infer 'supported' $^{210}Pb$ (Fig S1 a). $^{137}Cs$ and $^{241}Am$ were measured at 662keV and 59.5keV (Appleby et al., 1986), with absolute efficiencies determined by comparison with standards (Fig. S1 b). The constant rate of $^{210}Pb$ supply (CRS) model was used because slight departures from a monotonic declining trend in $^{210}Pb$ activity indicated variable sediment supply.
A composite age-depth model was produced using the 1963 peak in $^{137}Cs$ and $^{241}Am$ from radioactive fallout at 3.25 cm, as the $^{210}Pb$ placed this slightly earlier at 5 cm and so was corrected by 1.75 cm at this point in line with radiometric dating convention (Appleby, 2001) (Fig. 6). Estimated ages in the extrapolated part of the core (below 10 cm) are assumed to have the same error (± 18 years) as final measured sample in 1845 AD. The organic carbon mass accumulation rate (CMAR) was obtained by multiplying the dry mass accumulation rate (DMAR) by the
decimal of TOC (SI Fig. 3).

### 2.3 Geochemical analyses

### 2.3.1 Chlorophyll and carotenoid pigment analysis

Chlorophyll *a* (chl *a*) was measured spectrophotometrically against an extraction solvent blank, after extracting Whatman G/C filter residues in acetone overnight (Jeffrey and Humphrey, 1975). Filtered residues from Whatman GF/F (0.45 µm pore size) and freeze-dried sediments were extracted in an acetone, methanol, and water mixture (80:15:5), filtered using a 0.22 µm polytetrafluoroethylene (PTFE) filter, dried under $N_2$ gas and re-dissolved into injection solution, prior to analysis on an Agilent 1200 series high-performance liquid chromatography (HPLC) unit using a ODS Hypersill column (250x4.6mm; 5µm particle size) and a photo-diode array detector (350-750nm). Separation conditions are detailed in McGowan et al. (2012). Pigments were identified and quantified from the chromatograms by comparing retention times and spectral characteristics with calibration standards (DHI Denmark).

### 2.3.2 Bulk $\delta^{13}C$ and C/N analysis

$\delta^{13}C_{org}$ and C/N analyses were conducted on sediment core, soil and vegetation samples. Sediment core and soil samples were pre-treated in excess 5% HCl (hydrochloric acid), dried and homogenised with an agate pestle and mortar. Plants and bryophyte samples were placed in 5% HCl for 5 minutes to remove carbonates, checked for a visible reaction, washed in deionised $H_2O$, dried and ground to powder by freezer milling using liquid nitrogen. TOC, TN (from which we calculated C/N ratios) and $\delta^{13}C_{org}$ analyses were performed at the British Geological Survey online using a Costech ECS4010 elemental analyser (EA) coupled to a VG Triple Trap and a VG Optima dual-inlet mass spectrometer. $\delta^{13}C$ values were calibrated to the VPDB (Vienna Pee Dee Belemnite) scale using within-run laboratory standards (BROC2 and SOILB) calibrated against NBS-18, NBS-19 and NBS-22. Analytical precision of $\delta^{13}C_{org}$ was to within ±<0.1‰ (1 SD). C/N is presented as the weight ratio. To facilitate statistical analysis samples were grouped into aquatics, core sediments, terrestrial vascular plants and 'terrestrial other', with significant differences examined using one-way ANOVA. Post-hoc Tukey HSD was confirmed using Welch's t-test. An isotopic mixing model based on a Bayesian framework was developed using SIAR V4 (Stable Isotope Analysis in R) (Inger et al., 2010) in R (v3.5.2) (R Core Team, 2020) to couple catchment $\delta^{13}C_{org}$ and C/N ratio as source data with down-core measurements (grouped by sedimentary zone). Proportional boxplots were produced for both broad catchment groups (terrestrial other, vascular plant and aquatics) and at the finer species level (SI Fig. 4-7). Trophic enrichment factors were set at a mean of 1.63 ‰ and standard deviation of 0.63 ‰ for $\delta^{13}C$ (Inger et al., 2010), with a value of zero used for C/N ratio.

### 2.3.3 Lipid biomarker analysis

Lipid compounds were extracted from freeze dried and homogenised plant, benthic macroalgae, soil and sediment samples from Disko 2 using 15 mL HPLC grade 3:1 $CH_2Cl_2$:$CH_3OH$ and a MARS 5 microwave system (CEM Microwave Technology, UK). The total lipid extracts (TLEs) were dried under $N_2$, saponified using 6% KOH in $CH_3OH$, and the neutral compounds separated using hexane (after Pearson et al., 2011). Excess salts were removed from the neutral fraction using $CH_2Cl_2$ extracted ultrapure $H_2O$. The neutral fractions were derivatised using bis(trimethylsilyl)trifluoroacetamide (BSTFA) to form trimethylsilyl esters prior to analysis by gas chromatography mass spectrometry (GC-MS). Following the removal of the neutral compounds, the remaining

extract was acidified with HCL and the acid fraction subsequently separated using hexane. Prior to analysis by GC-MS the isolated acid fraction was esterified using BF3/MeOH and the fatty acid methyl esters (FAMEs) extracted using $C_6H_{14}$. Neutral and FAME samples were analysed using an Agilent 7890A GC coupled to a 5975C MS. Samples were injected in pulsed splitless mode at 280 ºC in $CH_2Cl_2$ using a HP DB5-MS (30m x 0.25 mm i.d: 0.25µm film thickness) column and with a temperature program, compound identification and quantification following Pearson et al. (2007). Diagnostic lipid ratios and equations were selected based on those which were most informative in the catchment study to interpret changes in source inputs (Fig. 5) and down-core changes in OM source and reactivity (Fig. 7b).

### 2.3.4 Compound-specific isotope analysis

Compound-specific $\delta^{13}C$ was obtained by injecting FAME acid fractions spiked with squalene (temperature 290ºC) in splitless mode into a Thermo Finnigan Trace GC coupled to a Thermo Finnigan Delta Plus XP isotope ratio mass spectrometer using a combustion interface (GC-C-IRMS) according to conditions in Huang et al. (2018). Samples were run in at least duplicate, with reproducibility achieved to at least ±0.5 ‰ (SD). An *n*-alkane standard was used as a within-run laboratory standard with known $\delta^{13}C$ values (Chiron, Norway) and daily combined isotopic reference samples from $C_{12}$ to $C_{32}$ (Indiana University). $\delta^{13}C$ values were relative to the VPDB (‰) and corrections were made for the additional C atoms introduced by $BF_3$-MeOH derivatisation (Boschker et al., 1999).

### 3 Results

### 3.1 Hydro-geomorphology and vegetation surveys of lake catchments

The highest elevation cirque lake (Disko 2) had the least imagery-derived catchment vegetation and the smallest catchment: lake area (CA:LA) ratio (Table 1, Fig. 2). The U-shaped valley bottom lake (Disko 1) with a moderately-sized CA:LA ratio had the highest proportion of imagery-derived vegetation cover. The lowest elevation valley-end lake (Disko 4) was located <2 km from to the coast and had moderate imagery-derived vegetation cover and the highest CA:LA ratio. All lakes had permanent glacial ice present in the catchment, but this coverage was greatest in the catchment of Disko 1 which is fed directly by the Lyngmarksbræen ice cap (Fig. 1c). Close to the lake margins, plant cover (assessed by vegetation transects) was highest at Disko 4 and lowest at Disko 2 (Table 1). Detailed vegetation composition surveys are available in Table S1.

### 3.2 Water chemistry

In all lakes, soluble reactive phosphorus (SRP) concentrations were higher in April than in August, with the highest levels in Disko 1 and lowest in Disko 2 (Table 2). Seasonal total phosphorus (TP) concentrations in Disko 1 were similar in April (67.9 µg $L^{-1}$) and August (67.8 µg $L^{-1}$), but declined in Disko 2 (from 37.5 µg $L^{-1}$ in April to 2.9 µg $L^{-1}$ in August) and Disko 4 (164.8 µg $L^{-1}$ in April to 2.9 µg $L^{-1}$ in August). Dissolved inorganic nitrogen (DIN) was present predominantly as nitrate ($NO_3$-N) which was highest in Disko 1 (0.05 µg $L^{-1}$ in August) and lowest in Disko 2 (0.01 µg $L^{-1}$ in April) whereas $NH_4^+$-N concentrations were below detection limits, except in August in Disko 1 and Disko 2 (11.54 µg $L^{-1}$ and 3.15 µg $L^{-1}$ and respectively). DOC was low in all lakes in both April and August; the highest values were recorded in April for Disko 4 (1.5 mg $L^{-1}$). Biomarker pigments in the

water column of all lakes in August (ice-free conditions) were characterised by high concentrations of pheophytin *b* (0.95 nmol L$^{-1}$), a chlorophyll *b* derivative from chlorophytes, euglenophytes and higher plants and lower concentrations of diatoxanthin from siliceous algae. Diatoxanthin concentrations were much higher in Disko 1 (0.63 nmol L$^{-1}$) than in the other lakes (Disko 2 – 0.05 nmol L$^{-1}$; Disko 4 0.08 nmol L$^{-1}$).

**3.3 TOC, C/N ratios and δ$^{13}$C values (bulk and compound specific) of modern samples**

TOC content varied between a maximum of 66.3% in herbaceous plants to 0.1% in soil and 0.5% in lake surface sediment (Fig. 3a). C/N ratios varied most in lichens (from 25 to nearly 200) and terrestrial plants (~15 to ~130) and were lowest in algae (9-17), aquatic macrophytes (9-36) and lake surface sediment samples (7.6-8.6) (Fig. 3b). δ$^{13}$C$_{org}$ ranged between −6.4‰ and −33.6 ‰ (Fig. 3c). Comparing the C/N ratio and δ$^{13}$C$_{org}$ values, most samples plotted across the ranges typical of C$_3$ land plants and lacustrine algae (Leng et al., 2012; Meyers and Teranes, 2001; Anderson et al., 2018), with one outlying aquatic macrophyte sample (Disko 2 *Potamogeton sp.* −6.4 ‰) and some lichen, moss and soil samples that had higher δ$^{13}$C$_{org}$ values (~ −20 to −23 ‰) (Fig. 4). Benthic macroalgae samples had δ$^{13}$C$_{org}$ values around −25 ‰. Lake sediment core samples from Disko 2 & 4 had low C/N ratios (7.5 to 18.4) and δ$^{13}$C$_{org}$ values ranged from −28 to −20 ‰. Statistical analyses demonstrated that C/N ratios of terrestrial vascular plants were significantly different from aquatics and lake sediments, and terrestrial 'other' also differed from lake sediments (Table 3). There was more overlap in δ$^{13}$C$_{org}$ and so terrestrial vascular plant measurements were only significantly different to core sediments and terrestrial 'other' (Table 3; Fig. 4). Compound-specific δ$^{13}$C$_{FAMEs}$ values tended to become progressively lower (more negative) for higher chain length fatty acids than shorter chains, compared with bulk δ$^{13}$C$_{org}$ (SI Fig. 3).

**3.4 Lipid distributions and ratios from the Disko 2 catchment**

Lipid histograms of *n*-alkane, *n*-alkanol and *n*-alkanoic acid distributions from Disko 2 (Fig. 5; SI 3) showed that *n*-alkane samples displayed an odd over even predominance whereas *n*-alkanols and *n*-alkanoic acids were even-dominated, as expected for naturally-derived samples (Bianchi and Canuel, 2011). Proportionately (given limitations of single ion filtering) *n*-alkanoic acids were the most abundant (mean 56.7%; range 11.7-98.8%) followed by *n*-alkanols (mean 31.5%; range 0.8-83.5%) and *n*-alkanes (mean 11.8%; range 0.5-59.0%). Differentiation between catchment samples was possible, for example among *n*-alkanes the algal benthic rock scrape sample was dominated by *n*-C$_{23}$, while for *Potamogeton* sp. and green moss, *n*-C$_{31}$ was dominant. Herbaceous plants like *Harrimanella hypnoides* tended to be dominated by *n*-C$_{29}$ or *n*-C$_{31}$ while dominant chain lengths were slightly shorter in *Salix arctica* (*n*-C$_{27}$).

Soil sample *n*-alkane distributions were either clearly terrestrially influenced (e.g. catchment summit soil, dominant in *n*-C$_{31}$) or more mixed (e.g. Plot C soil sample, taken close to the Disko 2 shoreline) and surface sediments also reflected mixed inputs with a slight odd predominance and an extended range. *n*-Alkanols could also be used to differentiate between samples, for example, dominance of *n*-C$_{26}$ in the algal benthic rock scrape but *n*-C$_{22}$ in *Potamogeton* sp. Higher plants were typically mid-chain dominant in *n*-alkanols (e.g. *Harrimanella hypnoides* with a maximum at either *n*-C$_{26}$ or *n*-C$_{22}$ and *Salix arctica n*-C$_{24}$), and the bimodal distribution of lake surface sediments suggesting multiple inputs. Although the algal benthic rock scrape and one *H. hypnoides* sample had similar *n*-C$_{26}$ alkanol dominance, *H. hypnoides* also has higher chain length compounds which the algal benthic rock scrape does not and the degree of dominance in the algal rock scrape (*n*-C$_{26}$ ca. 20%) is different

compared with *H. hypnoides* ($n$-$C_{26}$ ca. 9%), making these compositions distinguishable. The $n$-alkanoic acid $n$-$C_{16}$ was ubiquitous and abundant in all samples, but distributions still provided some differentiation, for example a secondary abundance in green moss around $n$-$C_{24}$, $n$-$C_{22}$ in the unidentified herbaceous plant sample and $n$-$C_{24}$ in the *Salix arctica* (leaf) and $n$-$C_{18}$ in the plot C soil sample.

Lipid ratios for catchment and surface sediment samples are provided in Table 4 with interpretation of source attribution described in Table S2. In this system the $n$-alkane ratios CPI 2 appears to be a good indicator of terrestrial plants, $TAR_{HC}$ is an excellent indicator of leafy vascular plants, $P_{WAX}$ indicates multiple terrestrial inputs and $n$-$C_{27}$ is a good indicator of key plant species like *Salix arctica* and *Chamerium latifolium*. Among $n$-alkanoic acids $CPI_T$ is a good indicator of aquatic macrophytes and mosses and $n$-$C_{30}$ of non-woody herbaceous terrestrial inputs. $n$-Alkanol indicator ratios are $n$-$C_{16}$ for aquatic macrophytes and $n$-$C_{24}$ for woody terrestrial plants. The ratio of brassicasterol/ total sterols is a clear indicator of algae and moss.

**3.5 Down core changes in bulk and compound-specific isotopic and geochemical parameters**

Stratigraphic zones determined using optimal sum of squares partitioning were from 2010 – 1833 AD (Zone A), 1833 – 1530 AD (Zone B) and 1530 – 1334 AD (Zone C). In zone C (1300-1530 ($\pm18$) AD) TOC generally declined from ~3 to 1.4 %, except for a peak to ~5.7 % around ~1440 ($\pm18$) AD and $\delta^{13}C_{org}$ ranged between $-20.3$ ‰ and $-24.1$ ‰ (Fig. 7a). C/N ratio ranged between 9 and 12 (reaching higher value ~1510 ($\pm18$) AD)), while β-carotene ranged between 1.1 and 33.3 nmoles pigment $g^{-1}$ TOC with a peak around ~1440 ($\pm18$) AD (Fig. 7a). Zone B (1530 – 1833 ($\pm18$) AD) proxies were relatively stable, except for an increase to ~5% TOC ~1640 ($\pm18$) AD, where $\delta^{13}C_{org}$ was high (~ $-20.2$ ‰) and β-carotene reached ~24.1 nmoles pigment $g^{-1}$ TOC (Fig. 7a). In zone B around ~1643 ($\pm18$) AD there were peaks in CPI 2 and decreases in $P_{WAX}$ and the $C_{26}$ $n$-alkanes, with increases in $TAR_{HC}$ around 1762 ($\pm18$) AD and ~1653/1798 ($\pm18$) AD for the $C_{30}$ $n$-alkanoic acids (Fig.7b).

The most pronounced changes in all geochemical proxies was in zone A (1833 ($\pm18$) – 2013 ($\pm2$) AD) which includes the ~1850 ($\pm18$) AD end of the LIA cooling period (Fig.7). TOC increased from <2.3 to 7.4 % by the top of the core, while $\delta^{13}C_{org}$ decreased from ~ $-23.5$ to $-28$ ‰ by ~1940 AD where it remained stable to ~2003 ($\pm2$) AD (Fig. 7a). $\delta^{13}C_{28:0}$ FAME also decreased from $-26.3$ ‰ in ~ 1895 AD to $-35.6$ ‰ by ~2003 ($\pm2$) AD (Fig. 7a). In contrast, C/N increased fluctuations ~1923 ($\pm6$) AD (12.4) followed by a steep decline to ~7.5 by ~1987 ($\pm3$) AD, while β-carotene increased 70-fold (from 0.9 nmoles pigment $g^{-1}$ TOC in 1906 ($\pm7$) AD to 70.4 nmoles pigment $g^{-1}$ by 2003 ($\pm2$) AD)  (Fig. 7a). In zone A $P_{WAX}$, $CPI_T$, $C_{16}$ $n$-alkanol and the ratio of brassicasterol/total sterols increased to the top of the core, whilst the $C_{27}$ $n$-alkane, $C_{30}$ $n$-alkanoic acid and $C_{24}$ $n$-alkanols declined, and CPI 2 and $TAR_{HC}$ were both generally stable until the uppermost sample (Fig. 7b). Both DMAR and CMAR were characterised by a clear pulsed increase to higher levels between ~1930 and 1960 AD (SI Fig. 3).

SIAR isotopic mixing model analysis linking the catchment with down-core $\delta^{13}C$ and C/N ratios revealed higher proportional densities (expressed in both histograms and boxplots) for vascular plants in group 1 (zone A 2013-1830 AD) than groups 2 (zone B 1830-1530 AD) and 3 (zone C 1530 -1300 AD) (Fig SI 4 & 5). At the species level although mixed, changes in *Betula nana*, *Chamerion latifolium*, Eriophorum, guano and soil explain the most variation in group 1 (zone A 2013-1830 AD) the most evidenced by proportional histograms and boxplots (Fig SI 5 & 6). For both groups 2 (zone B 1830-1530 AD) and 3 (zone C 1530-1300 AD), although mixed, the aquatic group is the most important with, benthic algae the greatest contributor.

**4 Discussion**

**4.1 How is catchment and aquatic carbon structured on Disko?**

The soil TOC content of the catchments was highly variable (Fig. 3a). Local decomposition mediated by temperature and moisture can regulate Arctic soil loss or retention, and litter quality (Wookey et al., 2009). In Arctic regions generally, patchy organic soils indicate faster recycling rates or specific local sources of OM controlled by the topographically varied dynamic glacio-fluvial geomorphic system (Conant et al., 2011; Castellano et al., 2015; Anderson et al., 2017). Vegetation changed with elevation, with the highest lake (Disko 2) catchment having more 'bare ground' and rock/gravel substrates (Table 1). The terrestrial plant species we observed are common on Disko (e.g. Callaghan et al., 2011; Daugbjerg, 2003) and other parts of West Greenland (Fredskild, 2000), but the plant communities were less species-rich than in other lowland areas of Disko (Table 1, Table SI 1) (Callaghan et al., 2011). The woody species *Betula nana* and *Juniperus communis* were uncommon, and only present in the Disko 4 catchment (Table S1) probably because it is more sheltered with better drained soils than the other sites, and located closer to the high diversity of the coast (Callaghan et al., 2011; Hollesen et al., 2015).

Differences in vegetation cover, soil development and glacial meltwaters appeared to influence water chemistry and aquatic production. DOC concentrations were low in all lakes ($\leq$1.5 mg L$^{-1}$, Table 2), reflecting limited soil maturity and development in all three catchments, as observed in high Arctic lakes with similar topographies (Michelutti et al., 2002). The DOC concentrations are much lower than other low Arctic sites in Alaska or Norway which have more vegetated catchments (McGowan et al., 2018). Sparser vegetation cover of the Disko 2 (13%) catchment compared with Disko 1 (39%) probably reduced organic matter leachates from soils or permafrost, reducing microbial-induced phosphorus leaching (Buckeridge et al., 2015). Hence, TP and SRP concentrations were lower in Disko 2 than 4 (Table 2) during the spring thaw period. The extensive wetlands in the Disko 1 and 4 catchments are typical of Arctic environments where waterlogging leads to terrestrial OM transfer, leading to slightly higher C/N ratios in Disko 4 compared with Disko 2 which is located in a cirque where bare earth is predominant (Hobbie et al., 2000) (Table 1; Fig. 1, 4).

Permanent ice/glaciers accounted for between 3.6 and 14% of catchment cover (Table 1), and glacial inputs are known to influence waterbodies via effects of seasonal meltwater release on lake water residence time and nutrient release from glacial melt (Anderson et al., 2017; Slemmons et al., 2013). Of the three lakes, Disko 1 stands out with the highest glacier cover as part of the Lyngmarksbræen ice cap is present in the catchment and so the water chemistry is different to Disko 2 and 4. High summer nutrient conditions in Disko 1 (Table 2: SRP 13.8 µg L$^{-1}$, TP 67.9 µg L$^{-1}$ and NH$_4^+$ 11.5 µg L$^{-1}$) are probably linked to the catchment glacier source, which can influence algal production and community composition (Slemmons and Saros, 2012). Glacier outwash into Disko 1 appears to profoundly affect phytoplankton composition, with the degradation products (pheophytin *b,* a chlorophyll *b* derivative) being most abundant in Disko 1, whereas chlorophyll *a* was dominant at other sites (Table 2). Chlorophytes also grow on glaciers (Williamson et al., 2018) and degrade into pheophytin *b* by UVR damage to algae (Leavitt et al., 2003), which can be released to downstream waterbodies via ice melt. High diatoxanthin from siliceous algae in Disko 1 can be explained by photo-degradation due to low DOC (high water clarity) and shallow depth (<5.9 m), without refuge from UVR (Sommaruga, 2001), while high chlorophyll *a* in Disko 4 can be explained by nutrient inputs from a large catchment with extensive plant cover (Table 1). Overall,

345     catchment characteristics such as glaciers, soil type and plant cover appear to be tightly coupled to water quality, giving the potential for allochthonous inputs to influence autochthonous production.

**4.2 Can catchment and in-lake composition of modern bulk OM discriminate between C sources and sinks?**

Despite the potential for terrestrial-aquatic linkages the bulk geochemical composition of catchment vegetation and soils was very different to the surface and down-core lake sediments (Fig. 3 & 4), suggesting that lake OM was either predominantly derived from in-lake/ autochthonous rather than catchment sources, or was extensively chemically altered prior to deposition in the lakes.  (Table 3). The low C/N ratios of surface sediments (range 7.6 to 8.6) also suggest an algal (low cellulose, aquatic) source rather than terrestrial plants with much higher values: lichen (30.0 – 196.0), herbaceous plants (12.2 – 133.6) and moss (35.2 – 127.9) (Fig. 3b) (Meyers and Teranes, 2001). C/N variation in lichens is likely as some can be $N_2$ fixing. Degradation of terrestrial vegetation by microbial and physical processes can result in substantial DOC loss from catchments (Moody et al., 2013). However, the low OM abundance in these catchments likely leads to low DOC fluxes, as evidenced by low lake water DOC (all ≤1.5 mg $L^{-1}$). The OM sedimented in lakes is a combination of autochthonous and allochthonous sourced C often with varying susceptibility to degradation (Catalán et al., 2013). The low C/N values are similar to values measured in lakes in the Low Arctic area of Sisimiut in south-west Greenland (Leng et al., 2012; Anderson et al., 2018).

Compared with C/N, for $\delta^{13}C_{org}$ there was more overlap in values between surface sediment and terrestrial samples, making differentiation more challenging (Fig. 3c) probably due to the competing effects of productivity and source in influencing C isotope composition (Hodell and Schelske, 1998; Meyers and Teranes, 2001). Despite the overlap we found the $\delta^{13}C_{org}$ of terrestrial vascular plants was statistically distinct from core sediments, but also from samples in the terrestrial 'other' category (bryophytes and lichens) (Table 3), suggesting that autochthonous algae, and not terrestrial vegetation are the main source of sedimentary C (Meyers and Teranes, 2001). In contrast to higher plants, bryophytes lack stomata and their isotope fractionation is closer to atmospheric $CO_2$ (Smith and Griffiths, 1996) (Hanson et al., 2014; Glime, 2007). On the other hand, variability in $\delta^{13}C_{org}$ of aquatic algae and macrophyte samples (−6.4 ‰ to −30.7 ‰) is expected to be high because aquatic plants can access C (and DIC) from different lake habitats which are in variable equilibrium with atmospheric $CO_2$. Higher values of $\delta^{13}C_{org}$ in these lake sediments indicate either greater contributions of algae, phytobenthos and macrophytes, or increases in productivity. The few low C/N ratios (approaching ~12 in some terrestrial specimens e.g. *Salix arctica, Chamerion latifolium*) are probably linked to new plant growth (with less lignin) or differences in sample response to inorganic C cleaning during acidification (Brodie et al., 2011). Low values in soils may be caused by soil algae which are common in Arctic areas (Pushkareva et al., 2016) and loss of carbon by diagenesis.

**4.3 Does the molecular composition of modern samples help distinguish C sources and sinks?**

Several molecular ratio indicators were particularly useful markers of terrestrial samples (see Table S3 for interpretation summary). For *n*-alkanes, higher CPI 2 values were a good indicator of leafy catchment plants (e.g. *Harrimanella hypnoides, Chamerion latifolium*) (Table 4) (Marzi et al., 1993), although they made a small contribution to the total organic carbon pool. The low *n*-alkane CPI 2 values in soil and surface sediment reflect both the possibility of degradation of any minor terrestrial sources and in sediments, and the predominant aquatic source in the lake sediments evidenced by C/N ratios (Meyers and Ishiwatari, 1993) (Fig. 4). Similarly, the *n*-

alkane $TAR_{HC}$ ratio, commonly used as an indicator of terrestrial (higher) to aquatic algae (lower) inputs (Bourbonniere and Meyers, 1996), was a robust indicator of the ericaceous shrub *Harrimanella hypnoides* and other herbaceous plants. The $TAR_{HC}$ ratio had moderate values in aquatic macrophytes such as *Potamogeton* sp., but only minor values in *Salix arctica*. Highest *n*-alkane $P_{WAX}$ index values (Zheng et al., 2007) were found in vascular plants such as *Harrimanella hypnoides*, catchment soil samples, the aquatic macrophyte *Potamogeton* sp. and green moss. Although the $P_{WAX}$ ratio in this system cannot distinguish among specific terrestrial inputs, the low contributions from the algal benthic rock scrape sample, make the index a good indicator of terrestrial versus aquatic inputs. $C_{30}$ *n*-alkanoic acid contributions were highest in in the *Harrimanella hypnoides* (2) sample and the Plot C soil sample, and so this ratio appears to be a good indicator of non-woody vascular plant and soil terrestrial inputs. The $C_{24}$ *n*-alkanol ratio was a good indicator of lignin rich higher plants like *Salix arctica* and *Chamerion latifolium*.

Other molecular ratio indicators were useful markers of aquatic algae, macrophytes, bryophytes and soil samples (see Table S3 for interpretation summary). The $CPI_T$ index for *n*-alkanoic acids was a good indicator of how degraded (or petrogenic) the C is (Matsuda and Koyama, 1977), with highest contributions in *Potamogeton* sp., black moss and the algal benthic rock scrape indicating high proportions of non-degraded C (Table 4). The $C_{16}$ *n*-alkanol was most abundant in surface sediment, the aquatic macrophyte *Potamogeton* sp. and Plot C soil, but can also be interpreted as a bulk productivity indicator (Cranwell, 1981). Brassicasterol is known to be present in samples containing cyanobacteria, algal mats, diatoms and Brassicaceae (Pearson et al., 2007; Volkman, 2003; Volkman et al., 1998; Rampen et al., 2010), but is commonly used in lakes as a diatom biomarker due to its high abundance (Nelson and Sachs, 2014). The markedly higher brassicasterol contribution in the Disko 2 surface sediments points to a pelagic algal source (mainly diatoms) of this compound, with the next highest contributions in the algal benthic rock scrape sample, moss samples and the Plot C soil sample. The $C_{27}$ ratio to total saturated *n*-alkanes was also a good indicator of some plants such as the *Salix arctica* leaf fragment and *Chamerion latifolium*. Combined, these markers not only distinguish terrestrial from aquatic, but also infer more subtle changes in the type of terrestrial cover.

**4.4 Evidence for changes in C cycling through the LIA and recent warming periods**

The dominant feature of the lake sediments across all catchments is the low C/N ratios (Fig. 4), suggesting that most sedimentary OM in these lakes is autochthonous in origin, corroborated by high brassicasterol (algal indicator), $C_{16}$ *n*-alkanols (macrophyte indicator) and *n*-alkanoic acid $CPI_T$ ratio (indicator of both young, fresh OM and shorter-chain algal source) in the uppermost sediments (Table 4). This is mirrored by declines after ~1750 AD in both long chain *n*-$C_{27}$ and *n*-$C_{30}$ total *n*-alkanoic acids ratios (Fig.7b) which highlight that proportionally shorter chain lengths indicative of aquatic algae increased (Cranwell et al., 1987). Dated sediments from Disko 2 therefore allow shifts in predominantly autochthonous C sources to be tracked through the LIA cooling period (LIA ~1300 AD to ~1850 AD; Miller et al., 2012; Matthews and Briffa, 2005) and subsequent warmer period after 1950 to understand how catchment mediated climate changes affect C cycling in this snowpack-fed cirque lake.

Aside from brief increases in % TOC and β-carotene around ~1450 AD and ~1650 AD, there was a major shift in these proxies after~1860 AD (corresponding to the end of the LIA), to the present day with a >10-fold increase in β-carotene (from ~1910 AD to recent maximum) indicating a major increase in phototrophic

production. One scenario is that the catchment thawed and the permanent ice cover (snowpack, cirque glacier) reduced, resulting in nutrients being released into the lake stimulating algal growth (Slemmons and Saros, 2012; Slemmons et al., 2013). Increases in the length of the ice-free period have been correlated with diatom responses in High Arctic locations (Michelutti et al., 2020; Griffiths et al., 2017). Although changes in the ice-free period have been associated with biological and chemical changes in Low Arctic lakes (Saros et al., 2019; Hobbs et al., 2010), it is rare to see the major directional shifts in Low Arctic biota in response to warming that are more commonly observed in High Arctic sites.

The high elevation of Disko 2 (575 m.a.s.l) and its limited catchment vegetation appear to suggest that the lake response to warming is simple, directional and fundamentally changed carbon cycling in the lake by responding in a two-stage, pulsed manner. This contrasts with many Low Arctic systems, where catchment vegetation is more established (Anderson et al., 2019), leading to more complex whole-lake responses to climate. The probable loss of a cirque glacier in the catchment after the LIA would have changed meltwater regimes, reducing lake water dilution and turbidity, making the lake more suited to algal production after ~1900 AD. Decreases in C/N ratio and increases in β-carotene after ~1900 AD (Meyers and Teranes, 2001) indicate increased algal production, similar to post-LIA increases in pigments in Baffin Island lake sediments (Florian, 2016; Florian et al., 2015; Michelutti et al., 2005). The decrease in $\delta^{13}C_{org}$ in the most recent sediments in Disko 2 is opposite to the expectation that higher productivity from nutrient enrichment leads to higher $\delta^{13}C_{org}$ (Hodell and Schelske, 1998). This decline of ~1.4 ‰ over the last ~50 years may be partly accounted for by the Suess effect from fossil fuel contamination (Verburg, 2007; Tans et al., 1979). However, most of the reduction in $\delta^{13}C_{org}$ probably results from a source change such as increased algae and macrophyte abundance. Combined, these geochemical markers show that following the end of the LIA cooling, algal production increased markedly in Disko 2, with this shift becoming more pronounced with the onset of recent warming and regional Disko glacier retreat (since ~1950 AD) (Yde and Knudsen, 2007; Hollesen et al., 2015; Box, 2002). The major two-step threshold shift since the end of the LIA (Fig. 7), is likely regulated by catchment processes associated with a glacier and/or permanent ice processes (e.g. nivation hollows, other periglacial processes) in the Disko 2 catchment, releasing N and P to the lake, shifting nutrient conditions and stimulating an algal response. Although 2013 surveys revealed only a small amount of permanent ice in this catchment, it is likely that in the LIA there was a larger cirque glacier in the catchment. Glaciers can respond not only to temperature but also precipitation which regulates their hydrological connectivity and subsequent release of N, P and C to lakes. During retreat, ice-entrained nutrition can be discharged and soil surfaces exposed, contributing to threshold shifts in algal species and abundance (Slemmons et al., 2017; Slemmons et al., 2013; Colombo et al., 2019). One possible source of N and P accumulation on glaciers is via aeolian transport of dust, which has many potential sources on Disko, an actively glaciated environment, with likely greater permanent ice cover in the LIA. Mechanistically, regional dust variability in Greenland is known to be a key determinant of sediment supply (Bullard and Mockford, 2018) and there is evidence that dust supplied P can stimulate Arctic lake algal response (Anderson et al., 2017). This is because dust is known to stimulate microbial and algal abundance on the glacier surface (Stibal et al., 2015), which subsequently becomes entrained in glacier ice and released to downstream lakes on retreat.

The timing of the most marked lacustrine response (Fig. 7) appears to be from the mid-20[th] century onwards, slightly later than the changes after the end of the LIA. Evidence for recent regional warming includes significant Greenland summit annual surface temperature increase between 1982 and 2011 (McGrath et al., 2013),

increased runoff from melt from the Greenland ice sheet in response to climate warming (Hanna et al., 2008), recent warming at west Greenland coastal weather stations between 1981 and 2011/12 (Hanna et al., 2012), significant increases in the Greenland Blocking Index (GBI) since 1981 in all seasons and annually (Hanna et al., 2016) and significant warming of the polar water layer in Disko Bay, responsible (in part) for the retreat of the Jakobshavn Isbrae at Ilulissat (Myers and Ribergaard, 2013). Although recent warming (last ~40 years) is known to be the most pronounced, it is challenging to interpret if warming was more pronounced in the middle 20$^{th}$ century than ~100 years earlier, when the LIA ended due to limitations from instrumental temperature records. Taking a palaeoclimate approach Overpeck et al. (1997) summarised multiple records suggesting that peak 20$^{th}$ century temperature in 1945 AD across the Arctic was around 1.2 °C more than in 1910. This suggests that mid-20$^{th}$ century warmth in the Arctic overall was probably greater than at the end of the LIA, but due to the spatial heterogeneity of these climate changes in different localities of the Arctic it is hard to compare these time periods specifically on Disko Island. However, these indicators are not direct temperature or climate proxies (Fig 7), but rather indicators of carbon cycling which are indirectly related to both catchment processes and climate, releasing nutrients and carbon and stimulating the recent algal response. Temperature is typically only one of many factors which drive changes in ecological thresholds in Greenland lakes (Anderson et al., 2018; Law et al., 2015; McGowan et al., 2018; Saros et al., 2019; Axford et al., 2013). Therefore the ecological response to regional warming is probably indirect through changes in catchment and shifts in the terrestrial carbon cycle, rather than solely a direct lake-water algal temperature response. Interestingly, both DMAR and CMAR increased in a pulsed manner between ~1930 and 1960 AD, suggesting processes in the Disko 2 catchment may have been particularly active during the mid-20$^{th}$ century, likely associated with glacier retreat or changes in catchment periglacial processes.

Although terrestrial OM was unlikely to be the dominant source of OM to Disko 2 sediments, there are indications that the type and magnitude of terrestrial OM fluxes have subtly varied over time and that recent changes in the catchment may be reflected in changes in the sediment.  For example, increases in the C/N ratio around ~1900 AD suggest catchment dieback, facilitated by warming during melt season exposure from reduced ice cover (Leng et al., 2012). The composition of catchment-derived C also shifted after ~1895 AD with a decline in $\delta^{13}C_{28:0}$ FAME from terrestrial plants, indicating a shift in terrestrial species with lower $\delta^{13}C_{28:0}$ values as the catchment thawed and nutrient availability increased. $\delta^{13}C_{FAME}$ values of Disko 2 catchment samples were lower (isotopically-depleted) in higher plants (e.g. *Harrimanella hypnoides* & *Chamerion latifolium*) (Fig. S3). Increases in *n*-alkane CPI 2 index between 1700 and 1600 AD probably reflect an earlier period of increased herbaceous plant input as a result of erosive freeze-thaw action in the catchment (Fig.7b). Despite some degradation in the OM, the CPI 2 value is consistently >2 down core which suggests *n*-alkane diagenesis is not a key process, with relatively fresh material potentially sourced from release of preserved OM from permafrost thaw in the catchment from climate warming (Schuur et al., 2009; Pautler et al., 2010). The peak in *n*-alkane TAR$_{HC}$ (index strongly biased towards the ericaceous shrub *Harrimanella hyponoides* and other herbaceous plants) in the 1760s is probably linked to catchment input of terrestrial material brought in during the LIA, which is probably partly old organic C, fluvially released from soils (Vonk et al., 2010). The steep topography surrounding Disko 2 is suited to the delivery of old OM, with visible seasonal thaw slope failure (Fig 2.a) confirmed by occasional gravel layers in other cores in this lake (Stevenson, 2017) which are a probable C delivery source (Lamoureux and Lafrenière, 2014). However, the *n*-alkane P$_{WAX}$, an indicator of waxy hydrocarbons which

are particularly abundant in terrestrial plants, was highest in the most recent ~70 years implying recent delivery of allochthonous material. Although the $TAR_{HC}$ and $P_{WAX}$ records disagree, they both suggest that the type of terrestrial material brought into the lake in the most recent period has changed and could be related to the sensitivity of $TAR_{HC}$ to *Harrimanella hypnoides* and other herbaceous plants, or their expansion (Table 4). The ratio of $n$-$C_{24}$ to total $n$-alkanols declined only in the uppermost samples (post~2003 AD), perhaps influenced by the most recent changes in catchment vegetation on Disko (Hollesen et al., 2015), or dilution by recent abundant short chain $n$-alkanols. Increases in $P_{WAX}$ in the most recent sediments (Fig. 7b) highlights (through comparison with modern samples, Fig. 4, Table S3) that there are some recent increases in terrestrial contributions (but of a different source to that explained by $TAR_{HC}$) contemporaneous with the recent increases in algal material (Fig. 7). Interestingly, despite limited vegetation cover in the Disko 2 catchment, SIAR isotopic mixing model analysis linking catchment with down-core C/N and $\delta^{13}C_{org}$ shows higher proportion densities for vascular plants in group 1 (zone A 2013-1830AD) than further down the core, supporting the idea that changes in terrestrial vegetation cover were contemporaneous with changes in the sediment (Fig. SI 4 and 5). At a finer scale, highly mixed *Betula nana*, *Chamerion latifolium*, Eriophorum, guano and soil explain the most variation in zone A, whereas in earlier sediments (zone B 1830-1530 AD and zone C 1530 -1300 AD) benthic algae explain the most variation in down-core C/N and $\delta^{13}C_{org}$ (Fig. SI 6 and 7). This supports the idea that changes in the catchment vegetation linked to warming probably also played a role in the major autochthonous response in the Disko 2 lake.

**4.5 Implications for C cycling in a coupled lake-catchment system**

The limited terrestrial vegetation and poorly developed soils on Disko Island, result in minor allochthonous inputs of C into lakes relative to autochthonous sources, especially in high elevation lakes. In systems where DOC fluxes from catchment soils are limited, this means in-lake autotrophic production is closely coupled and responsive to nutrient release from the catchment. Based on C/N and $\delta^{13}C_{org}$ the net carbon balance in these lakes on Disko Island is more autochthonous than other low Arctic lakes in coastal south-west Greenland where catchment vegetation is more developed (Anderson et al., 2018). Instead, Disko lakes seem to be similar in character to those on Baffin Island (Florian, 2016; Florian et al., 2015; Michelutti et al., 2002; Michelutti et al., 2005) owing to similarities in latitude (and hence seasonal variability), proximity to oceanic influence and geomorphology. The relatively minor DOC concentrations in lakes on Disko is also in marked contrast to more vegetated Arctic lake-catchment systems in Alaska and Sweden (McGowan et al., 2018). Despite limited terrestrial-aquatic C fluxes in Disko lakes, there is likely a substantial minerogenic input related to the steep topography and proximity to active glacial and periglacial processes, probably influencing sedimentation rate to a greater extent than in lakes in a less rugged terrain.

In-lake changes in C cycling also appear to be mediated by close coupling of these lakes to permanent ice in the catchments through nutrients, DOC and outwash from glacial and periglacial processes, and potentially also length of the ice-free period. From a lake metabolic balance (or C flux) perspective, autotrophic production increases with temperatures, and may be associated with the length of the ice-free period (Michelutti et al., 2020; Griffiths et al., 2017; Saros et al., 2019). Increases in temperature also induced modest changes in catchment C fluxes. Together, these changes probably contribute to a modest but regionally important enhanced C store in similar lakes across the region. Our results are consistent with changes to the Arctic C cycle more broadly across the Northern Hemisphere (McGuire et al., 2009; Post et al., 2009). This confirms that even in poorly vegetated

catchments, changes in the character of plant cover and, more recently, increases in algal production, have occurred coincident with climate-mediated catchment change, likely coupled to glacier and permafrost/soil nutrient and C cycling.

## 5 Conclusions

Our study demonstrates that Arctic lakes in catchments with limited vegetation and located at the transition between the Low-High Arctic boundary can respond rapidly to recent warming throughout the Anthropocene. This results in pronounced increases in phototrophic production, indicative of a major shift in autochthonous OC cycling. Specifically, we found:

    i.   C in lake sediments in sparsely vegetated catchments mostly derives from autochthonous algal sources
550         with minor terrestrial catchment carbon inputs, likely involved in stimulating recent change;

    ii.   low lake water DOC concentrations suggest limited terrestrial-aquatic C transfer in comparison to other Low Arctic areas (e.g Sweden, Alaska) with denser vegetation cover;

    iii.   ice and glacier retreat appeared to regulate lake nutrient inputs and subsequently phototrophic productivity, especially since the end of the LIA;

iv.   increases in the length of the ice-free period during recent warming may also help to explain a recent regime shift evidenced by marked increases in autochthonous algae.

Principally our findings illustrate how lakes with limited allochthonous C input respond more directly to climate-related changes in comparison to the catchments that are typically in more extensively vegetated low
Arctic systems. This is important because future anticipated warming will likely cause extensive, widespread major regime shifts in similar high-moderate elevation transitional Low-High Arctic catchment-lake systems where catchment buffering in poorly vegetated catchments is limited.

**Data availability**

Datasets are archived in PANGAEA as https://doi.org/10.1594/PANGAEA.927276
and https://doi.org/10.1594/PANGAEA.928314

**Author contribution**

MAS wrote the initial manuscript and carried out geochemical analyses. SMcG and GEA supervised nutrient, pigment and isotope preparation at U of N. EJP supervised lipid analyses at NU, MJL isotope analyses at BG, VJJ age-depth estimations at UCL and XH CSIA at CUG. JJB assisted with statistical analyses and summer 2013
fieldwork. MAS, EJP and EW carried out spring 2013 sediment coring. All authors revised and approved the manuscript.

**Competing interests**

The authors declare that they have no conflict of interest.

**Acknowledgements**

Mark Stevenson gratefully acknowledges the receipt of a NERC/ESRC studentship (ES/J500100/1). We acknowledge grants IP-1393-1113 & IP-1516-1114 from the NERC Isotope Geosciences laboratory (NIGL) for the analysis of $\delta^{13}C_{org}$ & C/N ratios on sediment, soil and plant samples. Lipid and water chemistry analyses were funded by the Freshwater Biological Association's 2015 Gilson Le Cren Memorial Award to Mark Stevenson. We thank Teresa Needham, Christopher Kendrick, Julie Swales, Ian Conway, Graham Morris, Bernard Bowler,

Paul Donohoe, Qingwei Song and Jiantao Xue for technical support. We acknowledge the support of Handong Yang for radiometric dating and the drafting of Fig 6a. Financial support for fieldwork was awarded via the INTERACT transnational access scheme (grant agreement No 262693) under the European Community's Seventh Framework Programme and UK RI NERC grant NE/K000276/1. Logistical support is acknowledged from University of Copenhagen Arktisk Station including Ole Stecher, Kjeld Mølgaard and Erik.

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

**Tables**

**Table 1: Key lake characteristics. Presented in descending elevation order.**

| | Disko 2 | | Disko 1 | | Disko 4 | |
|---|---|---|---|---|---|---|
| Lake coordinates | 69º23.342'N, 53º24.085' W | | 69º21.204'N, 53º29.421'W | | 69º17.841'N, 53º48.548'W | |
| Elevation (m a.s.l) | 575 | | 299 | | 214 | |
| Lake type | Cirque lake | | U-shaped valley bottom lake | | Valley-end lake | |
| Outflow type | Over scree covered shallow wetlands | | Wide braided wetland outflow – *more marked channel ~50 m d/s* | | Direct outflow | |
| Total catchment area (ha) | 358 | | 1,455 | | 1,829 | |
| Imagery derived vegetation (ha) | 46 | 13% | 569 | 39% | 497 | 27% |
| Imagery derived bare earth/rock (ha) | 284 | 79% | 650 | 45% | 1,250 | 68% |
| Imagery derived permanent ice/glacier (ha) | 17 | 5% | 203 | 14% | 67 | 3.6% |
| Vegetation survey derived total moss/lichen | 27.3% | | 37.5% | | 37.0% | |
| Vegetation survey derived total plants | 19.2% | | 32.7% | | 44.4% | |
| Vegetation survey derived total bare ground | 53.5% | | 29.8% | | 18.6% | |
| Catchment: lake area ratio | 45:1 | | 66:1 | | 122:1 | |
| Outflow type | Over scree covered shallow wetlands | | Wide braided wetland outflow – *more marked channel ~50 m downstream* | | Direct outflow | |


**Table 2: Selected water chemistry spot samples from Disko Island lakes in April and August 2013. Dashes denoting missing values, BDL = below detection limit.**

| Lake | Disko 2 | | Disko 1 | | Disko 4 | |
|---|---|---|---|---|---|---|
| 2013 sampling | April | August | April | August | April | August |
| SRP ($\mu$g L$^{-1}$) | 10.5 | 0.7 | 59.8 | 13.8 | 16.3 | 1.9 |
| TP ($\mu$g L$^{-1}$) | 37.5 | 2.9 | 67.9 | 67.8 | 164.8 | 2.9 |
| NO$^{3-}$ ($\mu$g L$^{-1}$) | 0.01 | 0.04 | 0.05 | 0.03 | 0.02 | 0.04 |
| NH$_4^+$ ($\mu$g L$^{-1}$) | BDL | 3.2 | BDL | 11.5 | BDL | BDL |
| Spectrophotometric Chl *a* ($\mu$g L$^{-1}$) | 0.9 | 0.7 | 1.0 | 0.4 | 0.1 | 1.0 |
| Total N (mg L$^{-1}$) | − | BDL | − | BDL | − | BDL |
| DOC (mg L$^{-1}$) | 1.0 | 0.6 | 0.6 | 0.9 | 1.5 | 0.8 |
| Diatoxanthin (nmol pigment L$^{-1}$) | − | 0.05 | − | 0.63 | − | 0.08 |
| Chlorophyll *a* (nmol pigment L$^{-1}$) | − | 0.55 | − | 0.46 | − | 0.85 |
| Pheophytin *b* (nmol pigment L$^{-1}$) | − | 0.33 | − | 0.95 | − | 0.35 |
| Date of collection | 19/04 | 03/08 | 17/04 | 02/08 | 21/04 | 07/08 |



**Table 3: Significant differences between pairings using parametric ANOVA post-hoc Tukey and non-parametric Mann-Whitney U**

| | ANOVA post-hoc Tukey | Man-Whitney U | Significant for both parametric and non-parametric tests? |
|---|---|---|---|
| | p adj | p-value | |
| C/N | | | |
| Core sediments vs aquatics | 0.48 | 0.01 | ✗ |
| Terrestrial 'other' vs aquatics | <0.01 | 0.31^ | ✗ |
| Terrestrial vascular plants vs aquatics | 0.01 | 0.03 | ✓ |
| Terrestrial 'other' vs core sediments | <0.001 | <0.001 | ✓ |
| Terrestrial vascular plants vs core sediments | <0.001 | <0.001 | ✓ |
| Terrestrial vascular plants vs terrestrial 'other' | 0.94 | 0.30^ | ✗ |
| $\delta^{13}C_{org}$ | | | |
| Core sediments vs aquatics | 0.51 | 0.74 | ✗ |
| Terrestrial 'other' vs aquatics | 0.15 | 1^ | ✗ |
| Terrestrial vascular plants vs aquatics | <0.001 | 0.46 | ✗ |
| Terrestrial 'other' vs core sediments | 0.22 | 0.03 | ✗ |
| Terrestrial vascular plants vs core sediments | <0.001 | <0.001 | ✓ |
| Terrestrial vascular plants vs terrestrial 'other' | <0.001 | <0.001 | ✓ |

Significant differences ($p < 0.05$) shaded

^Cannot compute exact p-value with ties

**Table 4: Selected lipid ratios for catchment samples from Disko 2 analysed using GC-MS.**

| Sample | TOC % | $n$-alkane ratios | | | | $n$-alkanoic acid (FAMEs) ratios | | $n$-alkanol ratios | | Sterols |
|---|---|---|---|---|---|---|---|---|---|---|
| | | CPI 2[1] | TAR HC[2] | $P_{WAX}$[3] | $n$-C$_{27}$/ total $n$-alkanes | CPI$_T$[4] | $n$-C$_{30}$ / total $n$-alkanoic acids | $n$-C$_{16}$ / total $n$-alkanols | $n$-C$_{24}$ total $n$-alkanols | (Brassicasterol /total sterols) *1000 |
| *Harrimanella hypnoides* (1) | 43.0 | 39.6 | 18899.2 | 1.2 | 14.4 | 43.2 | 0.7 | 0.1 | 9.8 | 6.4 |
| *Harrimanella hypnoides* (2) | 42.4 | 27.6 | 2980.3 | 1.0 | 5.5 | 15.3 | 4.2 | 0.7 | 16.3 | 16.3 |
| Unidentified herbaceous plant | 49.2 | 25.9 | 1523.1 | 1.1 | 10.8 | 24.3 | 0.6 | 0.3 | 8.5 | 7.1 |
| Black moss | 40.9 | 3.7 | 3.0 | 0.7 | 11.2 | 69.7 | 0.0 | 8.2 | 11.0 | 56.6 |
| *Potamogeton* sp. | 32.1 | 3.6 | 68.5 | 0.9 | 7.7 | 74.3 | 0.0 | 12.4 | 9.8 | 14.5 |
| Green moss | 34.9 | 5.4 | 8.0 | 0.9 | 10.3 | 32.4 | 0.1 | 0.1 | 26.2 | 44.2 |
| Algal benthic rock scrape | 24.5 | 3.5 | 6.2 | 0.5 | 7.7 | 40.6 | 0.0 | 0.8 | 25.5 | 57.8 |
| *Chamerion latifolium* | 33.1 | 16.4 | 129.9 | 0.7 | 26.0 | 27.5 | 0.7 | 0.0 | 29.3 | 3.5 |
| *Salix arctica* (leaf) | 47.0 | 18.5 | 19.6 | 0.9 | 31.3 | 17.3 | 0.0 | 0.2 | 42.2 | 0.4 |
| Catchment – Soil[5] | 0.3 | 5.2 | 357.7 | 1.1 | 11.1 | 15.1 | 0.2 | 2.0 | 21.6 | 20.4 |
| PLOT C – Soil[6] | 0.1 | 1.3 | 18.9 | 0.8 | 10.2 | 11.7 | 2.0 | 4.9 | 12.3 | 44.7 |
| Surface sediment | 7.4 | 2.4 | 5.1 | 0.8 | 6.0 | 24.2 | 0.1 | 16.7 | 10.8 | 608.7 |

[1]CPI 2 – Carbon Preference Index 2 (Marzi et al., 1993) = $((C_{23}+C_{25}+C_{27})+(C_{25}+C_{27}+C_{29}))/2*(C_{24}+C_{26}+C_{28})$

[2]TAR$_{HC}$ – Terrigenous to Aquatic Ratio (Bourbonniere and Meyers, 1996) = $(C_{27}+C_{29}+C_{31})/(C_{15}+C_{17}+C_{19})$

[3]$P_{WAX}$ – Index of waxy $n$-alkanes to total hydrocarbons (Zheng et al., 2007) = $(C_{27}+C_{29}+C_{31})/(C_{23}+C_{25}+C_{29}+C_{31})$

[4]CPI$_T$ – Carbon Preference Index (for the entire range) (Matsuda and Koyama, 1977) =
$0.5*((C_{12}+C_{14}+C_{16})+(C_{22}+C_{24}+C_{26}+C_{28}+C_{30}))+((C_{14}+C_{16}+C_{18})+(C_{24}+C_{26}+C_{28}+C_{30}+C_{32}))/((C_{13}+C_{15}+C_{17})+(C_{23}+C_{25}+C_{27}+C_{29}+C_{31}))$
Brassicasterol = 24-methylcholesta-5,22-dien-3β-ol
[5]from close to the highest point in the catchment 893 m (catchment summit)
[6]from close to shoreline (PLOT C)

**Figures**

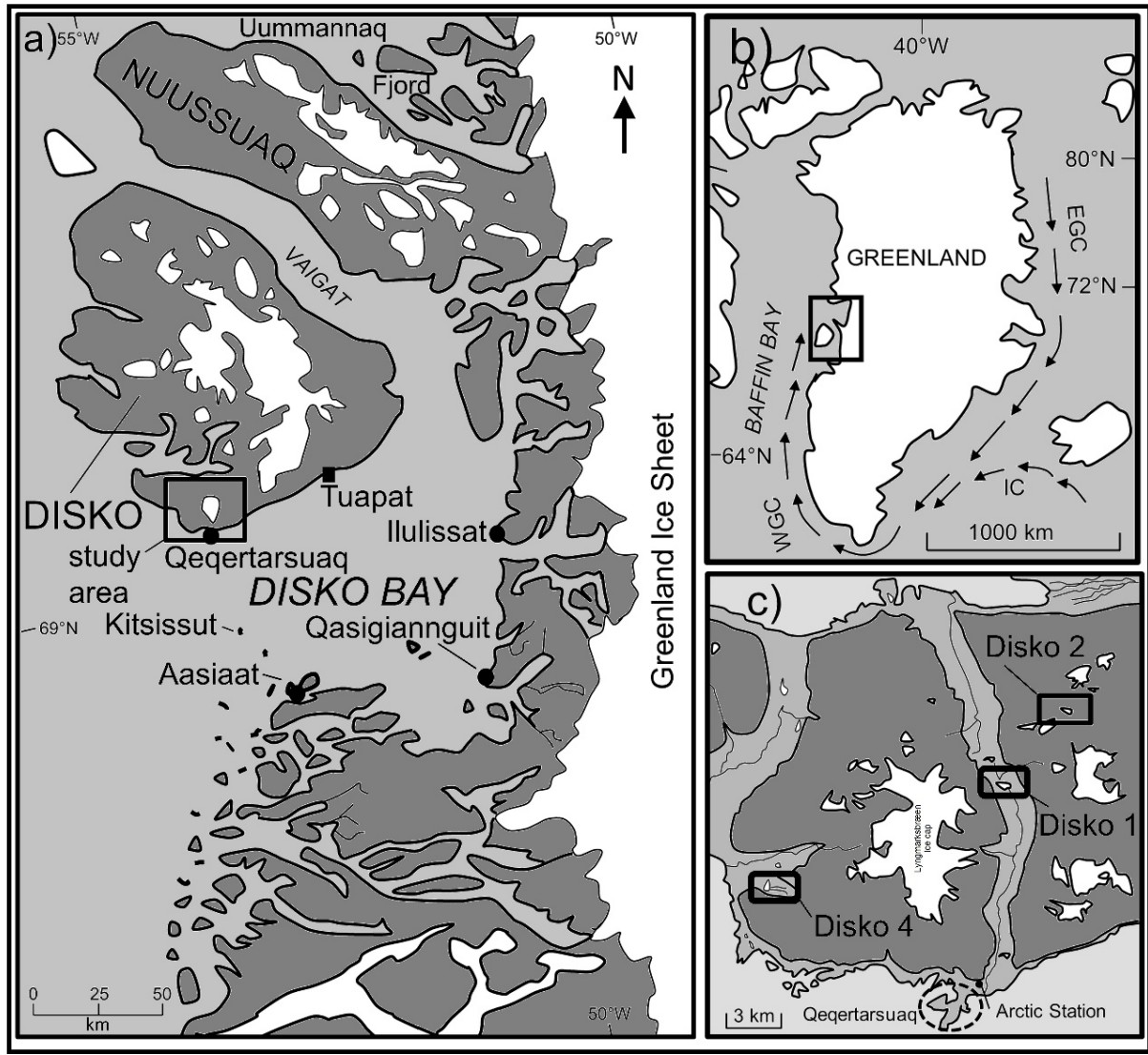

**Figure 1: a) Location of Disko Island and study area (indicated) in West Greenland. b) Location of Disko Island relative to Greenland. c) Location of study catchments to the north of Qeqertarsuaq and Disko Island. See Stevenson (2017) for local catchment maps.**

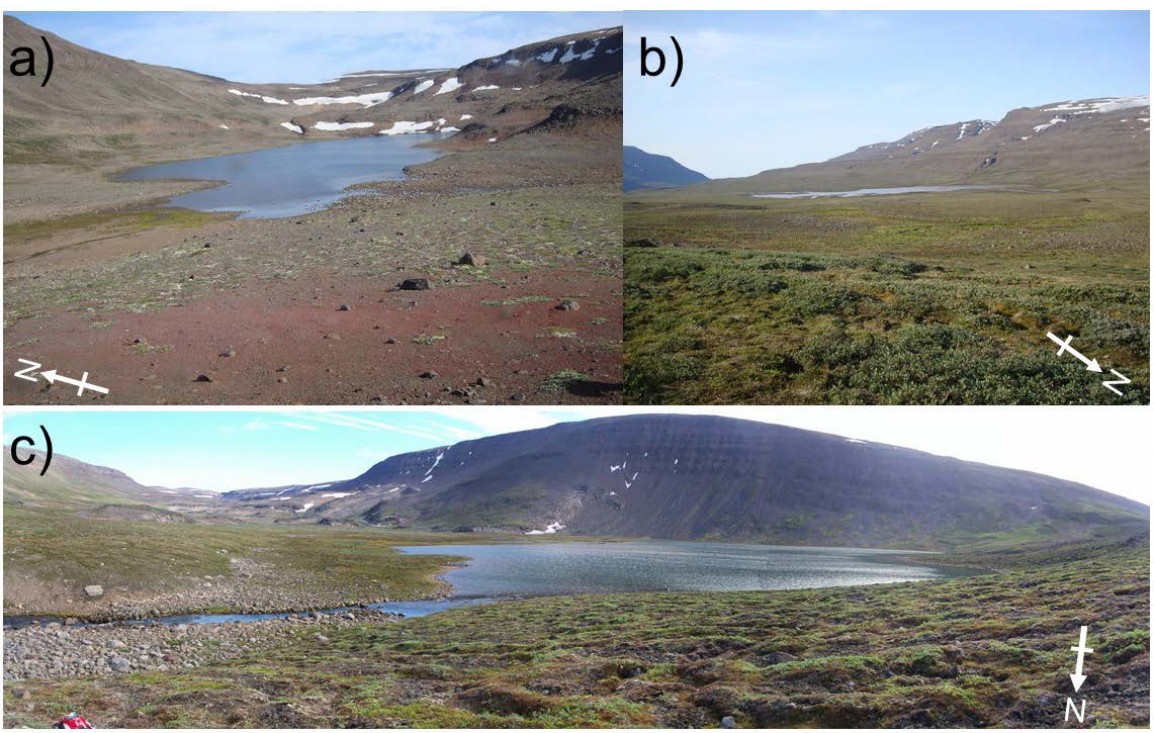

**Figure 2: Images of study catchments and lakes on Disko Island in August 2013: a) Lake Disko 2 (M.Stevenson); b) Lake Disko 1 (M.Stevenson); c) Lake Disko 4 (J.Bailey).**

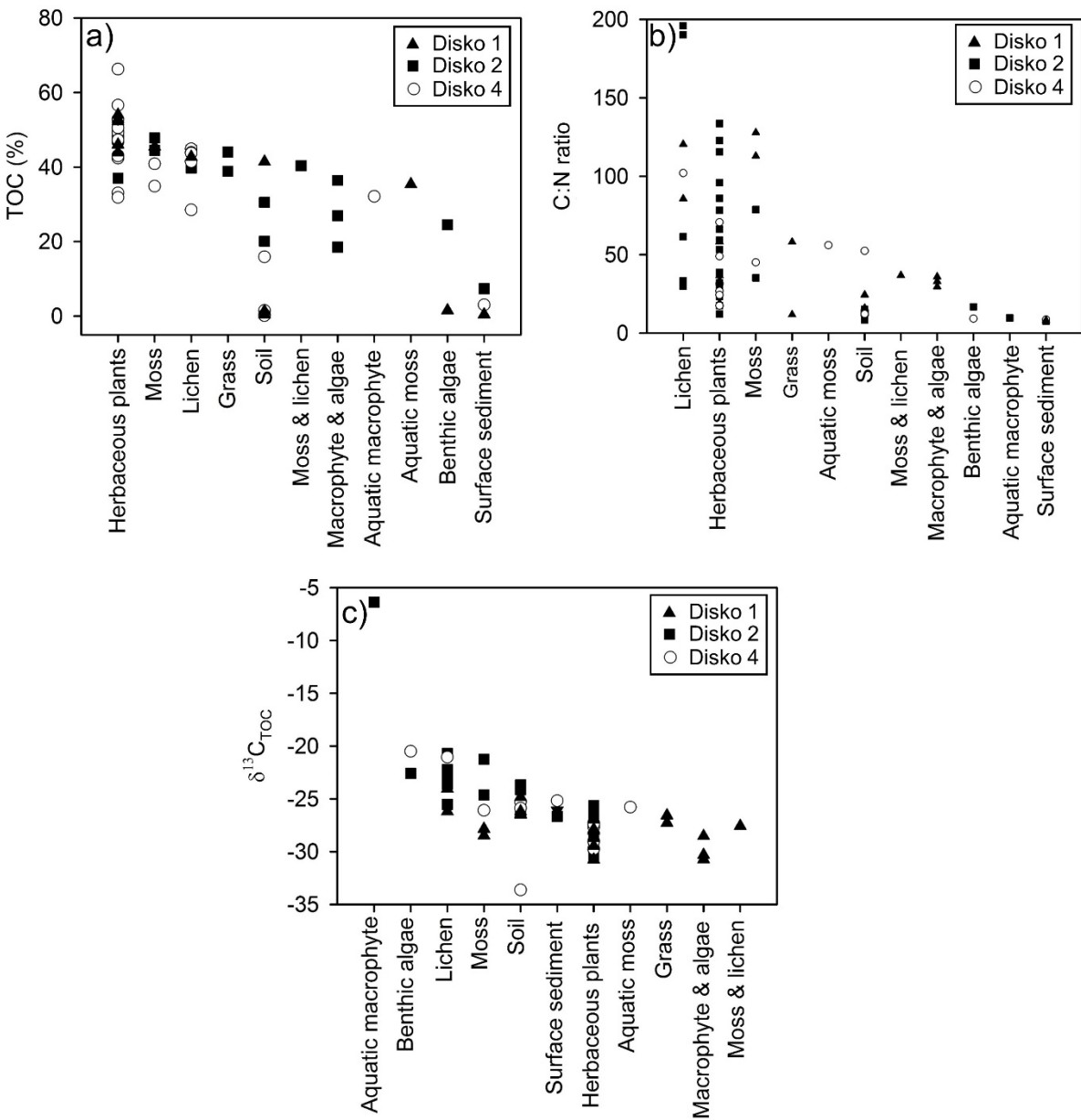

**Figure 3: a) TOC (%); b) C$_{org}$:N ratio; and c) δ$^{13}$C$_{org}$ plotted against sample type, separated by lake catchment.**

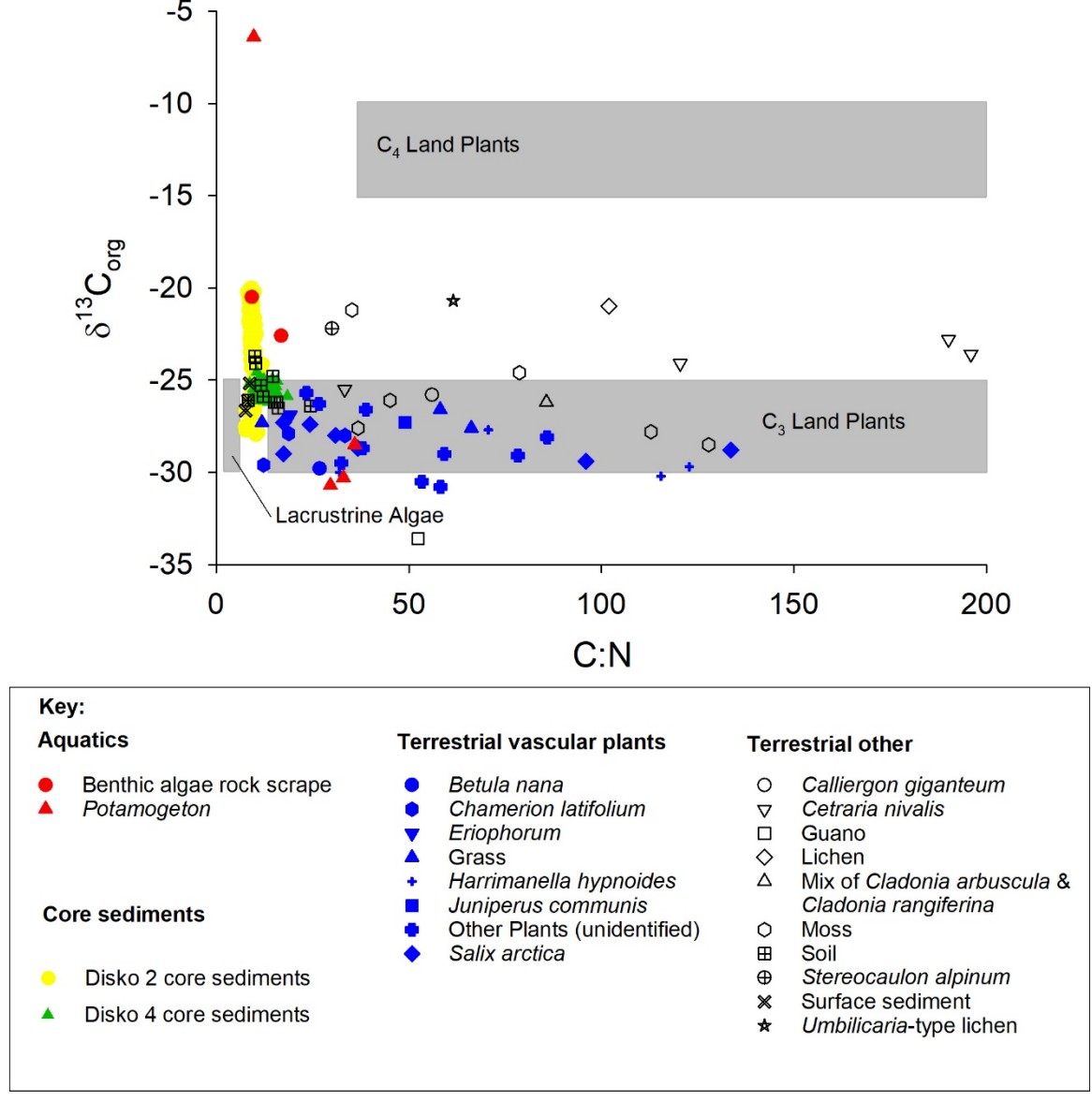

**Figure 4: C$_{org}$/N vs δ$^{13}$C$_{org}$ for catchment end-member samples and sediments from all three Disko catchment study areas. Grey shaded regions show expected zones for C$_4$ land plants, C$_3$ land plants and lacustrine algae according to Meyers (2003).**

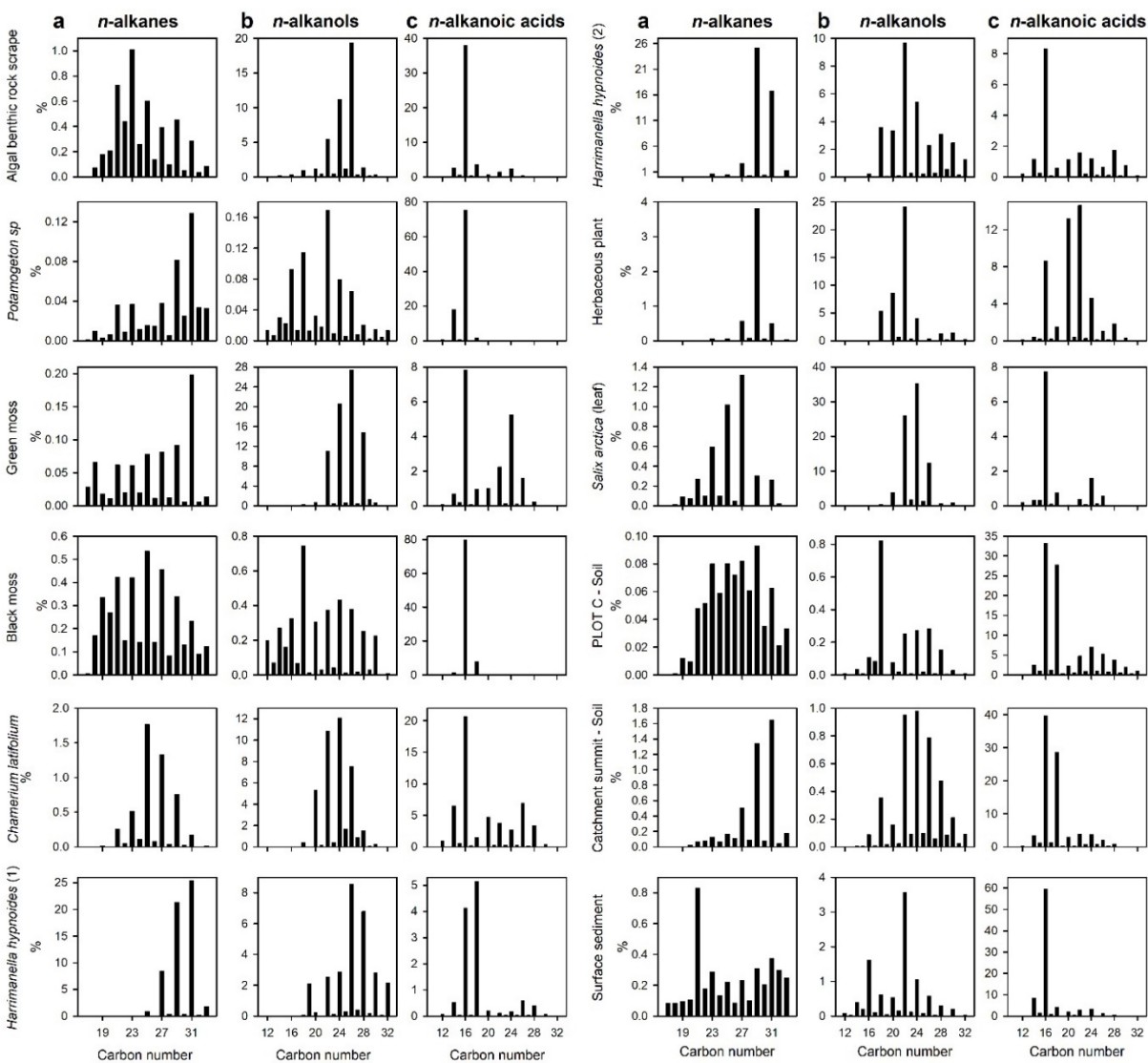

**Figure 5: Lipid distribution histograms for a)** *n*-alkanes, **b)** *n*-alkanols and **c)** *n*-alkanoic acids for catchment plant, soil and surface sediment samples from Disko 2 lake and catchment analysed using GC-MS. Expressed as relative proportion of total saturated *n*-alkanes, *n*-alkanols and *n*-alkanoic acids. Soil samples are taken from close to the Disko 2 shoreline (PLOT C) and close to the highest point in the catchment 893 m (catchment summit). Locations detailed in Stevenson (2017). Numbers (1) and (2) indicate replicates of separate samples from the Disko 2 catchment with the same identification.

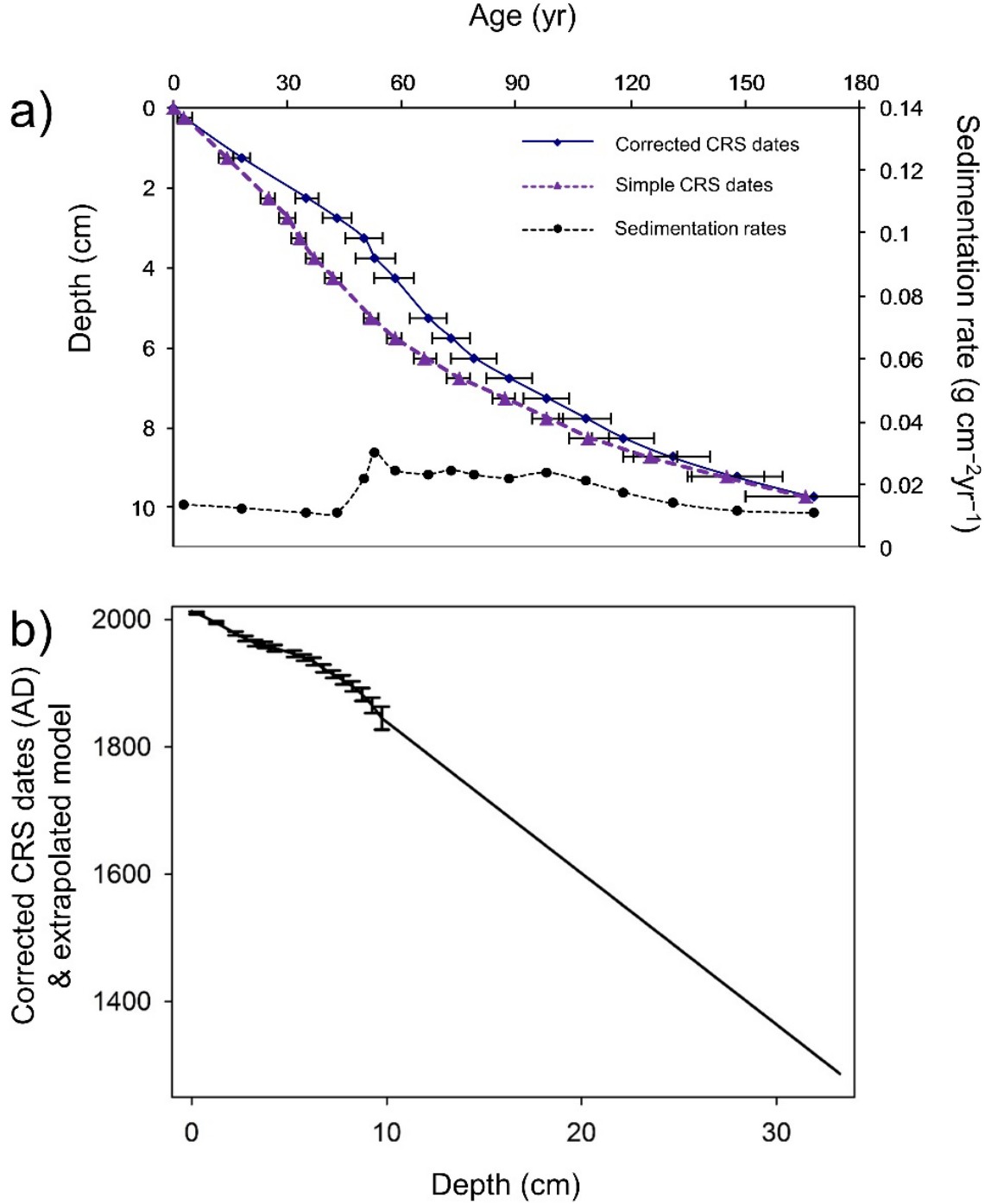

**Figure 6: a) Age depth models and sedimentation rates using corrected CRS model for the Disko 2 K1 core. The simple CRS (constant rate of supply) model derived from $^{210}$Pb dating only is indicated by purple triangles. The corrected CRS (constant rare of supply) model is derived from a $^{210}$Pb base, with manual splice corrections based on the $^{137}$Cs 1963 peak indicated by blue circles. b) Extrapolated model for Disko 2 K1 short core with linear extrapolation of dates between 1940 and 1845 AD used to extend the chronology to the base of the core. Error bars indicate measured $^{210}$Pb samples to 1σ standard error.**

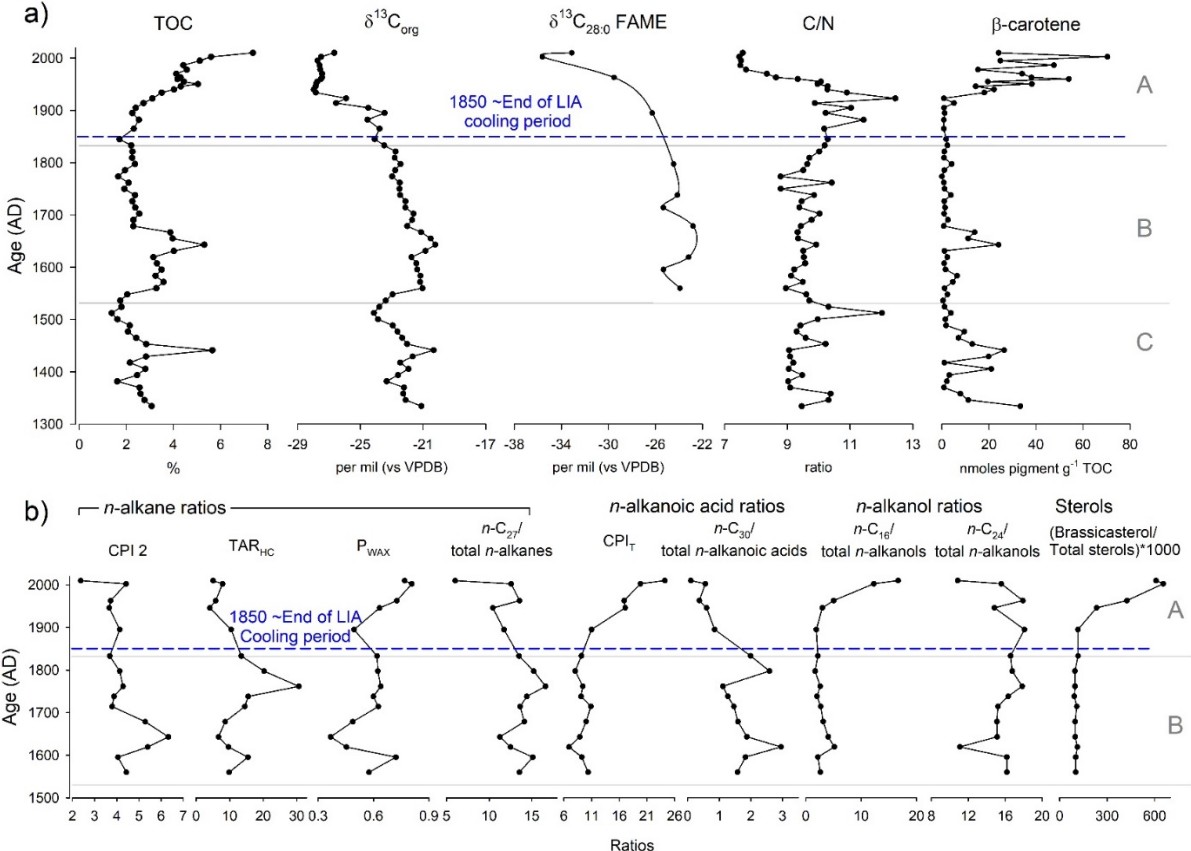

**Figure 7: a) Down-core bulk and compound-specific isotopic geochemical parameters from the Disko 2 (K1) short core, presented against modelled age (AD). b) Down-core lipid geochemical ratios and equations including CPI 2, TAR$_{HC}$, P$_{WAX}$, $n$-C$_{27}$/total $n$-alkanes, CPI$_T$, $n$-C$_{30}$/total $n$-alkanoic acids, $n$-C$_{16}$/total $n$-alkanols, $n$-C$_{24}$/total $n$-alkanols and (brassicasterol/total sterols)\*1000. For abbreviations see Table 4.**