# Peer review of "Anthropocene climate warming enhances autochthonous carbon cycling in an upland Arctic lake, Disko Island, West Greenland"

_Biogeosciences, 2020_

## Referee Comment (RC1) · Anonymous Referee #1 · 17 Nov 2020

This a solid contribution that documents the impact of recent warming on the productivity of — and flux of carbon to— arctic lakes. While I am not an organic geochemist and cannot comment on that aspect of the paper, the straightforward interpretation of the d13C, C/N, and pigment data clearly reveal an increase in the autochthonous input of organic matter to lacustrine sedimentary sequences after the end of the Little Ice Age.

I have a couple of questions regarding the limnologic response to the warming.

First, there appear to be a threshold response of C/N and d13C rather than a gradual response. WHat is the origin of this threshold response? Is it simply a build up of

nutrients in the water column that reaches some critical level that drives a lakes trophic system to increase substantially? Are there watershed filters at work that delay the response of a lake as recorded in the Disko 1 core?

Second, the main lacustrine response appears to take place in the middle of the 20th century, well after the end of the LIA. Again, is this reflecting a threshold temperature response by arctic lakes of regional warming? Or, was warming much more pronounced in the middle 20th century than 100 years earlier, when the LIA ended? Or, are there issues with the age model that could explain this difference.

Thirdly, how much would the plots of total C change when plotted as flux (mg/cmˆ2/yr-1 rather than %)? THe inflection in the age model might have a small effect on the shape of the C flux curve.

Lastly, what is the source of nutrients to these lakes. Mention is made of soil derived nutrients, but what about eolian accumulation of N and P on glacier surfaces that then are liberated to lake upon ice retreat.

Finally, the 137Cs data points are NOT clear on Fig.6

---

## Referee Comment (RC2) · Anonymous Referee #2 · 18 Nov 2020

This is a solid paleolimnological study. It combines several lines geochemical evidence to investigate changes in composition and concentration of organic matter. The MS is clearly written and it is scientifically sound. I have some comments that I hope authors find them positive and constructive.

Chronology. Have the chronology data been published elsewhere? Authors should show $^{210}Pb$ and $^{137}Cs$ activities, from which the chronological/sedimentation model was calculated and inferred. What is the correspondence between de $^{210}Pb$ and $^{137}Cs$ models?

Composition of organic matter, C and N isotopic data. Authors made an excellent effort in measuring samples of different nature and origin. In addition to what is presented in Fig. 4, authors should perform analysis in SIAR (https://maths.ucd.ie/~parnell_a/media/SIAR_For_Ecologists.pdf). Why didn't you try this tool if you probably have nice endmember information? This will improve the results and discussion of the MS.

Tile. The way the title is presented, it focuses rather on a methodological aspect and geographical location of the study. It would be better if authors can think of a title describing the paleoenvironmental process involved (i.e., warming and eutrophication), to make more attractive to other scientists.

---

## Author Comment (AC2) · 31 Dec 2020

This is a solid paleolimnological study. It combines several lines geochemical evidence to investigate changes in composition and concentration of organic matter. The MS is clearly written and it is scientifically sound. I have some comments that I hope authors find them positive and constructive.

Reply: Thank you for your positive comments on the manuscript's approach.

Chronology. Have the chronology data been published elsewhere? Authors should show 210Pb and 137Cs activities, from which the chronological/sedimentation model

was calculated and inferred. What is the correspondence between de 210Pb and 137Cs models?

Reply: We adjusted the 210Pb model based on 137Cs peak at a specific time point (1963 - atomic 'bomb' peak), to develop a composite CRS (continuous rate of supply) model. We can certainly provide more information in the supplementary information regarding plots of total 210Pb activity, unsupported 210Pb, and 137Cs and 241Am concentrations versus depth. This will highlight the good correspondence between simple 210Pb and adjusted 137Cs models, requiring only a 1.75 cm offset. The chronology has not been published elsewhere. This response also helps address questions concerning the dating model used from reviewer #1.

Composition of organic matter, C and N isotopic data. Authors made an excellent effort in measuring samples of different nature and origin. In addition to what is presented in Fig. 4, authors should perform analysis in SIAR (https://maths.ucd.ie/âĹijparnell_a/media/SIAR_For_Ecologists.pdf). Why didn't you try this tool if you probably have nice endmember information? This will improve the results and discussion of the MS.

Reply: Thank you for the positive feedback and the additional suggestion. We have explored the use of the SIAR tool in R setting down-core $\delta$13C and C/N data (grouped by sedimentary zone, Fig.7a of the manuscript) as consumer data and catchment $\delta$13C and C/N as source data. In our trial plot (Fig. 1) from the suggested modelling we can see that changes in vascular plants are able to explain a greater proportion of the variation in group 1 (2013-1830 AD), compared with groups 2 (1830-1530 AD) and 3 (1530-1300 AD) which have more mixed contributions. This provides confirmatory support to the notion that changes in terrestrial vegetation (driven by warming) in the catchment since the end of the LIA (group 1, 2013-1830 AD) are probably helping stimulate (or at least correlate with) the clearly identifiable recent response in autochthonous algae (evidenced by $\beta$-carotene in Fig. 7a of the manuscript).

Further analysis at the species level (not shown here, but we suggest could also be included in a revised supplementary information section) suggests that although mixed, changes in Betula nana, Chamerion latifolium, Eriophorum, guano and soil explain variation in group 1 (2013-1830 AD) the most. For both groups 2 (1830-1530 AD) and 3 (1530-1300 AD), although mixed, the aquatic group is the most important, of which benthic algae is the greatest contributor. We suggest to include plots deriving from the SIAR tool in the supplementary information section of a revised manuscript and make supporting reference in the relevant section of the discussion.

Title. The way the title is presented, it focuses rather on a methodological aspect and geographical location of the study. It would be better if authors can think of a title describing the paleoenvironmental process involved (i.e., warming and eutrophication), to make more attractive to other scientists.

Reply: Thank you for your comment. We suggest an alternative title suggestion which encompass the paleoenvironmental and environmental change processes involved:

Anthropocene climate warming enhances autochthonous carbon cycling in an upland Arctic lake Disko Island, West Greenland

This revised title should be more impactful and should help readers gain the best possible overview of the manuscripts content.

———————————————————

a)

**Proportion densities for group 1**
2013-1830 AD

b)

**Proportion densities for group 2**
1830-1530 AD

c)

**Proportion densities for group 3**
1530-1300 AD

**Fig. 1.** Trial proportion histograms produced in SIAR for a) group 1 (2013-1830 AD), b) group 2 (1830-1530 AD) and c) group 3 (1530-1300 AD).

---

## Author Response (AR1)

This a solid contribution that documents the impact of recent warming on the productivity of and flux of carbon to arctic lakes. While I am not an organic geochemist and cannot comment on that aspect of the paper, the straightforward interpretation of the d13C, C/N, and pigment data clearly reveal an increase in the autochthonous input of organic matter to lacustrine sedimentary sequences after the end of the Little Ice Age.

Reply: Thank you for your positive comments. We agree autochthonous production is very important in this system and so make this a central theme of our manuscript.

I have a couple of questions regarding the limnologic response to the warming. First, there appear to be a threshold response of C/N and d13C rather than a gradual response. WHat is the origin of this threshold response? Is it simply a build up of nutrients in the water column that reaches some critical level that drives a lakes trophic system to increase substantially? Are there watershed filters at work that delay the response of a lake as recorded in the Disko 1 core?

Reply: We think that the threshold response in the Disko 2 catchment is likely linked with glacier or permanent ice processes (nivation hollows, other periglacial processes etc.) in the cirque valley which can harbour and subsequently release N & P to the downstream lake. Although our surveys in 2013 revealed only a small amount of permanent ice in this catchment, we expect that in the Little Ice Age (LIA) there was a larger cirque glacier in the catchment. Glaciers can respond not only to temperature but also precipitation which regulates their hydrological connectivity and subsequent release of N and P. Certainly, delivery of nutrients can elicit a threshold response in algae shifting species composition which may be responsible for the pulsed increase in β-carotene and threshold response of C/N and $\delta^{13}$C. In a future revision we can include more comments on this topic.

Action: This comment has been addressed by adding text to section 4.4 (lines 447-455) of the discussion on the origin of the threshold response of C/N and $\delta^{13}$C linked to nutrients in the water column and catchment filters which may have delayed the lacustrine response.

Second, the main lacustrine response appears to take place in the middle of the 20th century, well after the end of the LIA. Again, is this reflecting a threshold temperature response by arctic lakes of regional warming? Or, was warming much more pronounced in the middle 20th century than 100 years earlier, when the LIA ended? Or, are there issues with the age model that could explain this difference. Thirdly, how much would the plots of total C change when plotted as flux (mg/cm^2/yr-1 rather than %)? THe inflection in the age model might have a small effect on the shape of the C flux curve.

Reply: These are excellent questions. Recent temperatures in western Greenland are generally accepted to be higher in the last few decades than at the end of the Little Ice Age (LIA). Evidence includes significant Greenland summit annual surface temperature increase between 1982 and 2011 (McGrath et al. 2013), increased runoff from melt from the Greenland ice sheet in response to climate warming (Hanna et al. 2008), recent warming at west Greenland coastal weather stations between 1981 and 2011/12 (Hanna et al. 2012), significant increases in the Greenland Blocking Index (GBI) since 1981 in all seasons and annually (Hanna et al. 2016) and significant warming of the polar water layer in Disko Bay, responsible (in part) for the retreat of the Jakobshavn Isbrae at Ilulissat (Myers and Ribergaard 2013). We would like to add more discussion on this linking with regionally important papers if we are invited to submit a revised version of the manuscript.

We therefore know that recent warming is the most pronounced but it is still difficult to know if warming was more pronounced in the middle 20th century than 100 years earlier, when the LIA ended as instrumental temperature records do not cover the full LIA. Taking a paleoclimate approach Overpeck et al. (1997) summarised multiple records suggesting that peak 20th century temperature in 1945 AD across the Arctic was around 1.2 °C more than in 1910. This suggests that mid-20th century warmth in the Arctic overall was probably greater than end of LIA, but due to the spatial heterogeneity of these climate changes in different localities of the Arctic we don't know the extent of the response specifically on Disko Island.

However, it is important to emphasise this manuscript does not include direct temperature or climate proxies, but rather indicators of carbon cycling which are indirectly related to both catchment processes and climate, releasing nutrients and carbon, stimulating the recent algal response.  A number of paleolimnological manuscripts with a Greenland focus have found that temperature is typically only one of many factors which drive changes in ecological thresholds (Anderson et al. 2018; Law, Anderson, and McGowan 2015; McGowan et al. 2018; Saros et al. 2019; Axford et al. 2013). Therefore we think the ecological response to regional warming is probably indirect though changes in catchment and shifts in the terrestrial carbon cycle, rather than just a direct lake-water algal response temperature on its own. The response to reviewer #2 concerning additional SIAR analysis in R provides additional evidence supporting this point.

Action: This comment has been addressed by adding text to section 4.4 of the discussion (lines 462-482) to better consider the timing of the main lacustrine response which appears to take place after the end of the LIA in the middle 20th century. The revised discussion also provides an assessment of when warming was most pronounced and discusses interrelationships between warming and changes in the catchment. The additional SIAR analysis discussed at the end of section 4.4 (lines 514-521) is also helpful to underscore links between catchment and the lake.

Future potential work for example, the quantification of alkenones would enable the quantification of temperatures (mid-20th Century vs end of LIA warmth), but would require additional time-consuming analyses (not completed) and setting of a different research question and therefore is most suited to a future manuscript.

We have confidence in our age-depth model as we developed it both using $^{210}$Pb and $^{137}$Cs dating techniques, adjusting the $^{210}$Pb model slightly (small offset of 1.75 cm) as best practice based on the known time markers of radioactive fallout of $^{137}$Cs isotopes in the northern hemisphere, supporting the model. Further detail on the recent dating model approach is also provided in the response to reviewer #2. We did look at organic carbon mass accumulation rate (in g OC cm$^{-2}$ yr$^{-1}$; I (Stevenson 2017)) and found that, while the recent threshold increases are even more apparent, they don't fundamentally change our interpretations. We could include additional plots as supplementary information if thought helpful.

Action: This comment has been addressed by adding text to section 4.4 of the discussion (lines 482-485) where we now consider total C plotted as flux, rather than just % and make reference to a new plot in the supplementary information (Figure S2) . We report that the presence of a pulse in CMAR and DMAR in the mid-20th century underscores the close link between the lake and catchment.

Lastly, what is the source of nutrients to these lakes. Mention is made of soil derived nutrients, but what about eolian accumulation of N and P on glacier surfaces that then are liberated to lake upon ice retreat.

Reply: Aeolian accumulation of N and P dusts on glacier surfaces is certainly a possible scenario. Such dust can be entrained from proglacial floodplains and deposited on the previous anticipated glacier/permanent ice expected in this catchment especially during the LIA. This can lead to microbial seeding on glacier surfaces, stimulating algal growth directly on the ice, with deposits rich in N and P entrained within the ice, subsequently released during glacier melt and resulting in increased C values. Disko is an active glaciated environment and so there is likely to be many such sources. We can comment more in the discussion section on this potential additional source of N and P. Additional references we propose to cite include Bullard and Mockford (2018) (importance of dust variability in Greenland for sediment supply), Anderson et al. (2017) (role of dust providing P to stimulate lake algal response) and Stibal et al. (2015) (evidence of rich glacier attached microbial mat being stimulated by nutrient harbouring deposited dust particles).

Action: This comment has been addressed by adding text to section 4.4 of the discussion (lines 455-461) where we add text and references which consider the possibility of aeolian accumulation on glacier surfaces and subsequent liberation during retreat.

Finally, the 137Cs data points are NOT clear on Fig.6

Reply: We did not include $^{137}$Cs data points in Fig.6 as we only included the model outputs. We can include these plots in the supplementary information section if invited to revise our manuscript.

Action: See new figure Fig. S1: a) $^{210}$Pb supported and unsupported activity plotted against core depth and b) $^{137}$Cs and $^{241}$Am activity plotted against core depth.

**References**

Anderson, N. J., M. J. Leng, C. L. Osburn, S. C. Fritz, A. C. Law, and S. McGowan. 2018. 'A landscape perspective of Holocene organic carbon cycling in coastal SW Greenland lake-catchments', *Quaternary Science Reviews*, 202: 98-108.

Anderson, N. John, Jasmine E. Saros, Joanna E. Bullard, Sean M. P. Cahoon, Suzanne McGowan, Elizabeth A. Bagshaw, Christopher D. Barry, Richard Bindler, Benjamin T. Burpee, Jonathan L. Carrivick, Rachel A. Fowler, Anthony D. Fox, Sherilyn C. Fritz, Madeleine E. Giles, Ladislav Hamerlik, Thomas Ingeman-Nielsen, Antonia C. Law, Sebastian H. Mernild, Robert M. Northington, Christopher L. Osburn, Sergi Pla-Rabès, Eric Post, Jon Telling, David A. Stroud, Erika J. Whiteford, Marian L. Yallop, and Jacob C. Yde. 2017. 'The Arctic in the Twenty-First Century: Changing Biogeochemical Linkages across a Paraglacial Landscape of Greenland', *Bioscience*, 67: 118-33.

Axford, Yarrow, Shanna Losee, Jason P. Briner, Donna R. Francis, Peter G. Langdon, and Ian R. Walker. 2013. 'Holocene temperature history at the western Greenland Ice Sheet margin reconstructed from lake sediments', *Quaternary Science Reviews*, 59: 87-100.

Bullard, Joanna E., and Tom Mockford. 2018. 'Seasonal and decadal variability of dust observations in the Kangerlussuaq area, west Greenland', *Arctic, Antarctic, and Alpine Research*, 50: S100011.

Hanna, Edward, Thomas E. Cropper, Richard J. Hall, and John Cappelen. 2016. 'Greenland Blocking Index 1851–2015: a regional climate change signal', *International Journal of Climatology*, 36: 4847-61.

Hanna, Edward, Philippe Huybrechts, Konrad Steffen, John Cappelen, Russell Huff, Christopher Shuman, Tristram Irvine-Fynn, Stephen Wise, and Michael Griffiths. 2008. 'Increased Runoff from Melt from the Greenland Ice Sheet: A Response to Global Warming', *Journal of Climate*, 21: 331-41.

Hanna, Edward, Sebastian H. Mernild, John Cappelen, and Konrad Steffen. 2012. 'Recent warming in Greenland in a long-term instrumental (1881–2012) climatic context: I. Evaluation of surface air temperature records', *Environmental Research Letters*, 7: 045404.

Law, A. C., N. J. Anderson, and S. McGowan. 2015. 'Spatial and temporal variability of lake ontogeny in south-western Greenland', *Quaternary Science Reviews*, 126: 1-16.

McGowan, Suzanne, N. John Anderson, Mary E. Edwards, Emma Hopla, Viv Jones, Pete G. Langdon, Antonia Law, Nadia Solovieva, Simon Turner, Maarten van Hardenbroek, Erika J Whiteford, and Emma Wiik. 2018. 'Vegetation transitions drive the autotrophy–heterotrophy balance in Arctic lakes', *Limnology and Oceanography Letters*, 3: 246-55.

McGrath, Daniel, William Colgan, Nicolas Bayou, Atsuhiro Muto, and Konrad Steffen. 2013. 'Recent warming at Summit, Greenland: Global context and implications', *Geophysical Research Letters*, 40: 2091-96.

Myers, Paul G., and Mads H. Ribergaard. 2013. 'Warming of the Polar Water Layer in Disko Bay and Potential Impact on Jakobshavn Isbrae', *Journal of Physical Oceanography*, 43: 2629-40.

Overpeck, J., K. Hughen, D. Hardy, R. Bradley, R. Case, M. Douglas, B. Finney, K. Gajewski, G. Jacoby, A. Jennings, S. Lamoureux, A. Lasca, G. MacDonald, J. Moore, M. Retelle, S. Smith, A. Wolfe, and G. Zielinski. 1997. 'Arctic Environmental Change of the Last Four Centuries', *Science*, 278: 1251-56.

Saros, Jasmine E., Nicholas John Anderson, Stephen Juggins, Suzanne McGowan, Jacob C. Yde, Jon Telling, Joanna E. Bullard, Marian L. Yallop, Adam J. Heathcote, Benjamin T. Burpee, Rachel A. Fowler, Christopher D. Barry, Robert M. Northington, Christopher L. Osburn, Sergi Pla-Rabes, Sebastian H. Mernild, Erika J. Whiteford, M. Grace Andrews, Jeffrey T. Kerby, and Eric Post. 2019. 'Arctic climate shifts drive rapid ecosystem responses across the West Greenland landscape', *Environmental Research Letters*, 14: 074027.

Stevenson, Mark Andrew. 2017. 'Carbon cycling in Arctic lakes: sedimentary biomarker reconstructions from Disko Island, West Greenland', University of Nottingham.

Stibal, Marek, Erkin Gözdereliler, Karen A. Cameron, Jason E. Box, Ian T. Stevens, Jarishma K. Gokul, Morten Schostag, Jakub D. Zarsky, Arwyn Edwards, Tristram D. L. Irvine-Fynn, and Carsten S. Jacobsen. 2015. 'Microbial abundance in surface ice on the Greenland Ice Sheet', *Frontiers in Microbiology*, 6.

**Anonymous Referee #2

This is a solid paleolimnological study. It combines several lines geochemical evidence to investigate changes in composition and concentration of organic matter. The MS is clearly written and it is scientifically sound. I have some comments that I hope authors find them positive and constructive.

Reply: Thank you for your positive comments on the manuscript's approach.

Chronology. Have the chronology data been published elsewhere? Authors should show 210Pb and 137Cs activities, from which the chronological/sedimentation model was calculated and inferred. What is the correspondence between de 210Pb and 137Cs models?

Reply: We adjusted the $^{210}$Pb model based on $^{137}$Cs peak at a specific time point (1963 - atomic 'bomb' peak), to develop a composite CRS (continuous rate of supply) model. We can certainly provide more information in the supplementary information regarding plots of total $^{210}$Pb activity, unsupported $^{210}$Pb, and $^{137}$Cs and $^{241}$Am concentrations versus depth. This will highlight the good correspondence between simple $^{210}$Pb and adjusted $^{137}$Cs models, requiring only a 1.75 cm offset. The chronology has not been published elsewhere. This response also helps address questions concerning the dating model used from reviewer #1.

Action: See new figure Fig. S1: a) $^{210}$Pb supported and unsupported activity plotted against core depth and b) $^{137}$Cs and $^{241}$Am activity plotted against core depth. Also see lines 53-154 (methods section 2.2 – core chronology) in the tracked changes document which now refers to the 1.75 cm offset.

Composition of organic matter, C and N isotopic data. Authors made an excellent effort in measuring samples of different nature and origin. In addition to what is presented in Fig. 4, authors should perform analysis in SIAR (https://maths.ucd.ie/~parnell_a/media/SIAR_For_Ecologists.pdf). Why didn't you try this tool if you probably have nice endmember information? This will improve the results and discussion of the MS.

Reply: Thank you for the positive feedback and the additional suggestion. We have explored the use of the SIAR tool in R setting down-core δ$^{13}$C and C/N data (grouped by sedimentary zone, Fig.7a of the manuscript) as consumer data and catchment δ$^{13}$C and C/N as source data. In our trial plot (Fig. 1) from the suggested modelling we can see that changes in vascular plants are able to explain a greater proportion of the variation in group 1 (2013-1830 AD), compared with groups 2 (1830-1530 AD) and 3 (1530-1300 AD) which have more mixed contributions. This provides confirmatory support to the notion that changes in terrestrial vegetation (driven by warming) in the catchment since the end of the LIA (group 1, 2013-1830 AD) are probably helping stimulate (or at least correlate with) the clearly identifiable recent response in autochthonous algae (evidenced by β-carotene in Fig. 7a of the manuscript).

Further analysis at the species level (not shown here, but we suggest could also be included in a revised supplementary information section) suggests that although mixed, changes in *Betula nana*, *Chamerion latifolium*, *Eriophorum*, guano and soil explain variation in group 1 (2013-1830 AD) the most. For both groups 2 (1830-1530 AD) and 3 (1530-1300 AD), although mixed, the aquatic group is the most important, of which benthic algae is the greatest contributor. We suggest to include plots deriving from the SIAR tool in the supplementary information section of a revised manuscript and make supporting reference in the relevant section of the discussion.

[Figure]

Figure 1: Trial population histograms produced in SIAR for a) group 1 (2013-1830 AD), b) group 2 (1830-1530 AD) and c) group 3 (1530-1300 AD).

Action: We have applied the SIAR tool as described above to both group and species level data. Use of the SIAR tool in R is now introduced in lines 181-184 of the tracked changes document (see section 2.3.2 of the methods) and detailed plots are provided in the supplements (Fig. S4 – S7). It is also now mentioned in results section lines 301-308 and discussion section lines 514-521. Based on the SIAR analysis strengthening our understanding that recent changes in terrestrial vegetation are reflected in the recent sediments we have added "We also demonstrate recent changes in catchment terrestrial vegetation cover contributed to the autochthonous response" to the abstract (lines 42-43) and "…likely involved in stimulating recent change" to the abstract (line 553).

Title. The way the title is presented, it focuses rather on a methodological aspect and geographical location of the study. It would be better if authors can think of a title describing the paleoenvironmental process involved (i.e., warming and eutrophication), to make more attractive to other scientists.

Response: Thank you for your comment.  We suggest an alternative title suggestion which encompass the paleoenvironmental and environmental change processes involved:

Anthropocene climate warming enhances autochthonous carbon cycling in an upland Arctic lake Disko Island, West Greenland

This revised title should be more impactful and should help readers gain the best possible overview of the manuscripts content.

Action: The title has been amended to the version suggested above.